# Circular RNA repertoires are associated with evolutionarily young transposable elements

**Franziska Gruhl[1,2†], Peggy Janich[2,3‡], Henrik Kaessmann[4]\*, David Gatfield[2]\***

[1]SIB Swiss Institute of Bioinformatics, Lausanne, Switzerland; [2]Center for Integrative Genomics, University of Lausanne, Lausanne, Switzerland; [3]Krebsforschung Schweiz, Bern, Switzerland; [4]Center for Molecular Biology of Heidelberg University (ZMBH), DKFZ-ZMBH Alliance, Heidelberg, Germany

**Abstract** Circular RNAs (circRNAs) are found across eukaryotes and can function in post-transcriptional gene regulation. Their biogenesis through a circle-forming backsplicing reaction is facilitated by reverse-complementary repetitive sequences promoting pre-mRNA folding. Orthologous genes from which circRNAs arise, overall contain more strongly conserved splice sites and exons than other genes, yet it remains unclear to what extent this conservation reflects purifying selection acting on the circRNAs themselves. Our analyses of circRNA repertoires from five species representing three mammalian lineages (marsupials, eutherians: rodents, primates) reveal that surprisingly few circRNAs arise from orthologous exonic loci across all species. Even the circRNAs from orthologous loci are associated with young, recently active and species-specific transposable elements, rather than with common, ancient transposon integration events. These observations suggest that many circRNAs emerged convergently during evolution – as a byproduct of splicing in orthologs prone to transposon insertion. Overall, our findings argue against widespread functional circRNA conservation.

**\*For correspondence:**
h.kaessmann@zmbh.uni-heidelberg.de (HK);
david.gatfield@unil.ch (DG)

**Present address:** †SIB Swiss Institute of Bioinformatics, Lausanne, Switzerland; ‡Krebsforschung Schweiz, Bern, Switzerland

**Competing interests:** The authors declare that no competing interests exist.

## Introduction

First described more than 40 years ago, circular RNAs (circRNAs) were originally perceived as a curiosity of gene expression, yet they have gained significant prominence over the last decade (reviewed in *Kristensen et al., 2019*; *Patop et al., 2019*). Large-scale sequencing efforts have led to the identification of thousands of individual circRNAs with specific expression patterns and, in some cases, specific functions (*Conn et al., 2015*; *Du et al., 2016*; *Hansen et al., 2013*; *Piwecka et al., 2017*). CircRNA biogenesis involves so-called 'backsplicing', in which an exon's 3' splice site is ligated onto an upstream 5' splice site of an exon on the same RNA molecule (rather than downstream, as in conventional splicing). Backsplicing occurs co-transcriptionally and is guided by the canonical splicing machinery (*Guo et al., 2014*; *Ashwal-Fluss et al., 2014*; *Starke et al., 2015*). It can be facilitated by complementary, repetitive sequences in the flanking introns (*Dubin et al., 1995*; *Jeck et al., 2013*; *Ashwal-Fluss et al., 2014*; *Zhang et al., 2014*; *Liang and Wilusz, 2014*; *Ivanov et al., 2015*). Through intramolecular base-pairing and folding, the resulting hairpin-like structures can augment backsplicing over the competing, regular forward-splicing reaction. Backsplicing seems to be rather inefficient in most cases, as judged by the low circRNA expression levels found in many tissues. For example, it has been estimated that about 60% of circRNAs exhibit expression levels of less than 1 FPKM (fragments per kilobase per million reads mapped) – a commonly applied cut-off below which genes are usually considered to not be robustly expressed (*Guo et al., 2014*). Due to their circular structure, circRNAs are protected from the activity of cellular exonucleases, which is thought to favour their accumulation to detectable steady-state levels and,

together with the cell's proliferation history, presumably contributes to their complex spatiotemporal expression patterns (*Alhasan et al., 2016*; *Memczak et al., 2013*; *Bachmayr-Heyda et al., 2015*). Overall higher circRNA abundances have been reported for neuronal tissues (*Westholm et al., 2014*; *Gruner et al., 2016*; *Rybak-Wolf et al., 2015*) and during ageing (*Gruner et al., 2016*; *Xu et al., 2018*; *Cortés-López et al., 2018*).

All eukaryotes (protists, fungi, plants, animals) produce circRNAs (*Wang et al., 2014*). Moreover, it has been reported that circRNAs are frequently generated from orthologous genomic regions across species such as mouse, pig, and human (*Rybak-Wolf et al., 2015*; *Venø et al., 2015*), and that their splice sites have elevated conservation scores (*You et al., 2015*). In these studies, circRNA coordinates were transferred between species to identify 'conserved' circRNAs. However, the analyses did not distinguish between potential selective constraints actually acting on the circRNAs themselves, from those preserving canonical splicing features of genes in which they are formed (termed 'parental genes' in the following). Moreover, even though long introns containing reverse complement sequences (RVCs) appear to be a conserved feature of circRNA parental genes (*Zhang et al., 2014*; *Rybak-Wolf et al., 2015*), the rapid evolutionary changes occurring on the actual repeat sequences present a considerable obstacle to a thorough evolutionary understanding. Finally, concrete examples for experimentally validated, functionally conserved circRNAs are still rather scarce. At least in part, the reason may lie in the difficulty to specifically target circular vs. linear transcript isoforms in loss-of-function experiments; only recently, novel dedicated tools for such experiments have been developed (*Li et al., 2021*). Currently, however, the prevalence of functional circRNA conservation remains overall unclear.

Here, we set out to investigate the origins and evolution of circRNAs; to this end, we generated a comprehensive set of circRNA-enriched RNA sequencing (RNA-seq) data from five mammalian species and three organs. Our analyses unveil that circRNAs are typically generated from a distinct class of genes that share characteristic structural and sequence features. Notably, we discovered that circRNAs are flanked by species-specific and recently active transposable elements (TEs). Our findings support a model according to which the integration of TEs is preferred in introns of genes with similar genomic properties, thus facilitating circRNA formation as a byproduct of splicing around the same exons of orthologous genes across different species. Together, our work suggests that most circRNAs – even when occurring in orthologs of multiple species and comprising the same exons – may nevertheless not trace back to common ancestral circRNAs but have rather emerged convergently during evolution, facilitated by independent TE insertion events.

## Results

### A comprehensive circRNA dataset across five mammalian species

To explore the origins and evolution of circRNAs, we generated paired-end RNA-seq data for three organs (liver, cerebellum, testis) in five species (grey short-tailed opossum, mouse, rat, rhesus macaque, human) representing three mammalian lineages with different divergence times (marsupials; eutherians: rodents, primates) (*Figure 1A*). For optimal cross-species comparability, all organ samples originated from young, sexually mature male individuals; we used biological triplicates (*Supplementary file 1*), with the exception of human liver (single sample) and rhesus macaque cerebellum (duplicates). From the RNA extracted from each sample, we generated two types of libraries; that is, with and without prior treatment of the RNA with the exoribonuclease RNase R. This strategy allowed us to enrich for circRNAs (in libraries with RNase R treatment) and to calculate the actual enrichment factors (from the ratio with/without RNase R treatment). Using a custom pipeline that took into account RNase R enrichment and other factors to remove likely false-positives and low expression noise (see Materials and methods and *Supplementary file 2*), we then identified circRNAs from backsplice junction (BSJ) reads, estimated circRNA steady-state abundances, and reconstructed their isoforms (*Supplementary file 3*, *Figure 1—figure supplement 1*, *Figure 1—figure supplement 2*).

In total, following rigorous filtering, we identified 1535 circRNAs in opossum, 1484 in mouse, 2038 in rat, 3300 in rhesus macaque, and 4491 circRNAs in human, with overall higher numbers in cerebellum, followed by testis and liver (*Figure 1A*, *Supplementary file 4*). Identified circRNAs were

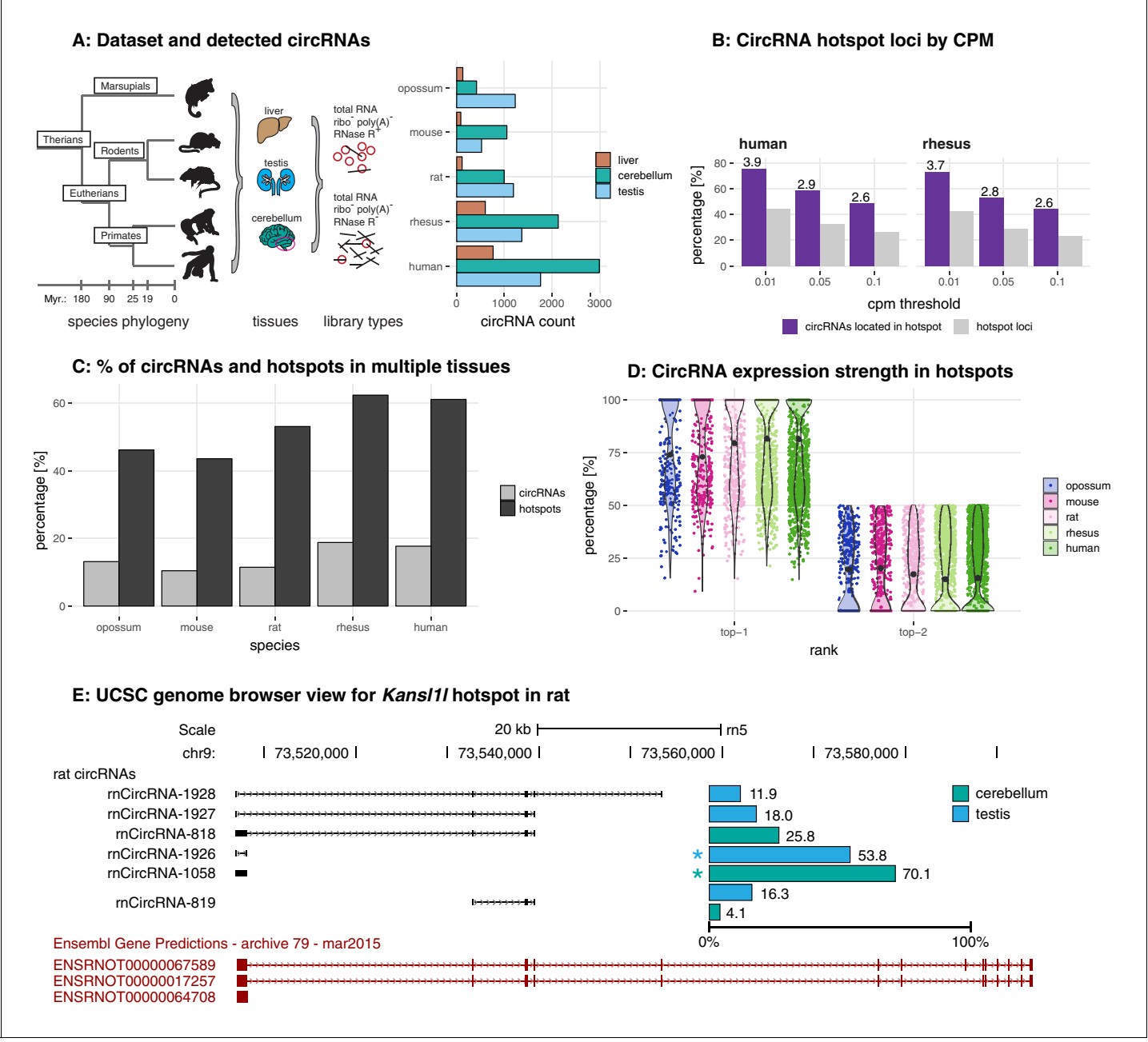

**Figure 1.** Study design, samples, datasets, and characterisation of circRNA properties and hotspots. (**A**) Phylogenetic tree of species analysed in this study and detected circRNAs. CircRNAs were identified and analysed in five mammalian species (opossum, mouse, rat, rhesus macaque, human) and three organs (liver, cerebellum, testis). Each sample was split and one half treated with RNase R to enrich BSJs. A dataset of high confidence circRNAs was established, based on the enrichment of BSJs in RNase R-treated over untreated samples. To the right of the panel, the total number of circRNAs for each species in liver (brown), cerebellum (green), and testis (blue) is shown. (**B**) CircRNA hotspot loci by CPM (human and rhesus macaque). The graph shows, in grey, the proportion (%) of circRNA loci that qualify as hotspots and, in purple, the proportion (%) of circRNAs that originate from such hotspots, at three different CPM thresholds (0.01, 0.05, 0.1). The average number of circRNAs per hotspot is indicated above the purple bars. (**C**) Number of circRNA hotspot loci found in multiple tissues. The graph shows the proportion (%) of circRNAs (light grey) and of hotspots (dark grey) that are present in at least two tissues. (**D**) Contribution of top-1 and top-2 expressed circRNAs to overall circRNA expression from hotspots. The plot shows the contribution (%) that the two most highly expressed circRNAs (indicated as top-1 and top-2) make to the total circRNA expression from a given hotspot. For each plot, the median is indicated with a grey point. (**E**) Example of the *Kansl1l* hotspot in rat. The proportion (%) for each detected circRNA within the hotspot and tissue (cerebellum = green, testis = blue) are shown. The strongest circRNA is indicated by an asterisk. rnCircRNA-819 is expressed in testis and cerebellum.

The online version of this article includes the following figure supplement(s) for figure 1:

*Figure 1 continued on next page*

**Figure supplement 1.** Overview of the reconstruction pipeline.
**Figure supplement 2.** Mapping summary of RNA-seq reads.
**Figure supplement 3.** General circRNA properties.
**Figure supplement 4.** CircRNA hotspot loci by CPM (opossum, mouse, rat).

generally small in size, overlapped with protein-coding exons, frequently detectable only in one of the tissues, and were flanked by long introns (*Figure 1—figure supplement 3*).

## The identification of circRNA heterogeneity and hotspot frequency is determined by sequencing depth and detection thresholds

Many genes give rise to multiple, distinct circRNAs (*Venø et al., 2015*). Such 'circRNA hotspots' are of interest as they may be enriched for genomic features that drive circRNA biogenesis. A previous study defined hotspots as genomic loci that produced at least ten structurally different, yet overlapping circRNAs (*Venø et al., 2015*). Reaching a specific number of detectable circRNA species for a given locus (e.g. 10 distinct circRNAs, as in the cited example) is likely strongly dependent on overall sequencing depth and on the CPM (counts per million) detection cut-off that is applied. We therefore compared circRNA hotspots identified at different CPM values (0.1, 0.05, and 0.01 CPM); moreover, to capture in a comprehensive fashion the phenomenon that multiple circRNAs can be generated from a gene, we considered genomic loci already as hotspots if they produced a minimum of two different, overlapping circRNAs at the applied CPM threshold. As expected, the number of hotspots – and the number of individual circRNAs that they give rise to – depend on the chosen CPM threshold (*Figure 1B* for human and rhesus macaque data; *Figure 1—figure supplement 4* for other species). Thus, at 0.1 CPM only 16–27% of all detected circRNA-generating loci are classified as hotspots. Decreasing the stringency to 0.01 CPM increases the proportion of hotspot loci to 32–45%. At the same time, the fraction of circRNAs that originate from hotspots (rather than from non-hotspot loci) increases from 34–49% (0.1 CPM) to 59–76% (0.01 CPM), and the number of circRNAs per hotspot increases from 2 to 6. Together, these analyses show that with lower CPM thresholds, the number of distinct circRNAs that become detectable per locus increases substantially; the number of detectable individual circRNA-generating loci increases as well, yet this effect is overall smaller. Furthermore, we observed that in many cases the same hotspots produces circRNAs across multiple organs (*Figure 1C*), with typically one predominant circRNA expressed per organ (*Figure 1D*). The *Kansl1l* hotspot locus is a representative example: it is a hotspot in rat, where it produces six different circRNAs (*Figure 1E*). It is also a hotspot in all other species and produces 8, 5, 7, and 6 different circRNAs in opossum, mouse, rhesus macaque and human, respectively (data not shown).

Overall, we concluded that the expression levels of many circRNAs are low. Increasing the sensitivity of detection (i.e. lowering CPM thresholds) led to a substantial gain in the detectability of additional, low-expressed circRNA species, but less so of additional circRNA-generating genomic loci. These findings raised the question whether many of the circRNAs that can be identified reflected a form of gene expression noise that occurred preferentially at hotspot loci, rather than functional transcriptome diversity.

## CircRNAs formed in orthologous loci across species preferentially comprise constitutive exons

We therefore sought to assess the selective preservation – and hence potential functionality – of circRNAs. For each gene, we first collapsed circRNA coordinates to identify the maximal genomic locus from which circRNAs can be produced (*Figure 2A*). In total, we annotated 5428 circRNA loci across all species (*Figure 2A*). The majority of loci are species-specific (4103 loci; corresponding to 75.6% of all annotated loci); there are only comparatively few instances where circRNAs arise from orthologous loci in the different species (i.e. from loci that share orthologous exons in corresponding 1:1 orthologous genes; *Figure 2A*). For example, only 260 orthologous loci (4.8% of all loci) give rise to circRNAs in all five species (*Figure 2A*). A considerable proportion of these shared loci also correspond to circRNA hotspots (opossum: 28.0%, mouse: 43.6%, rat: 53.0%, rhesus macaque: 46.2%,

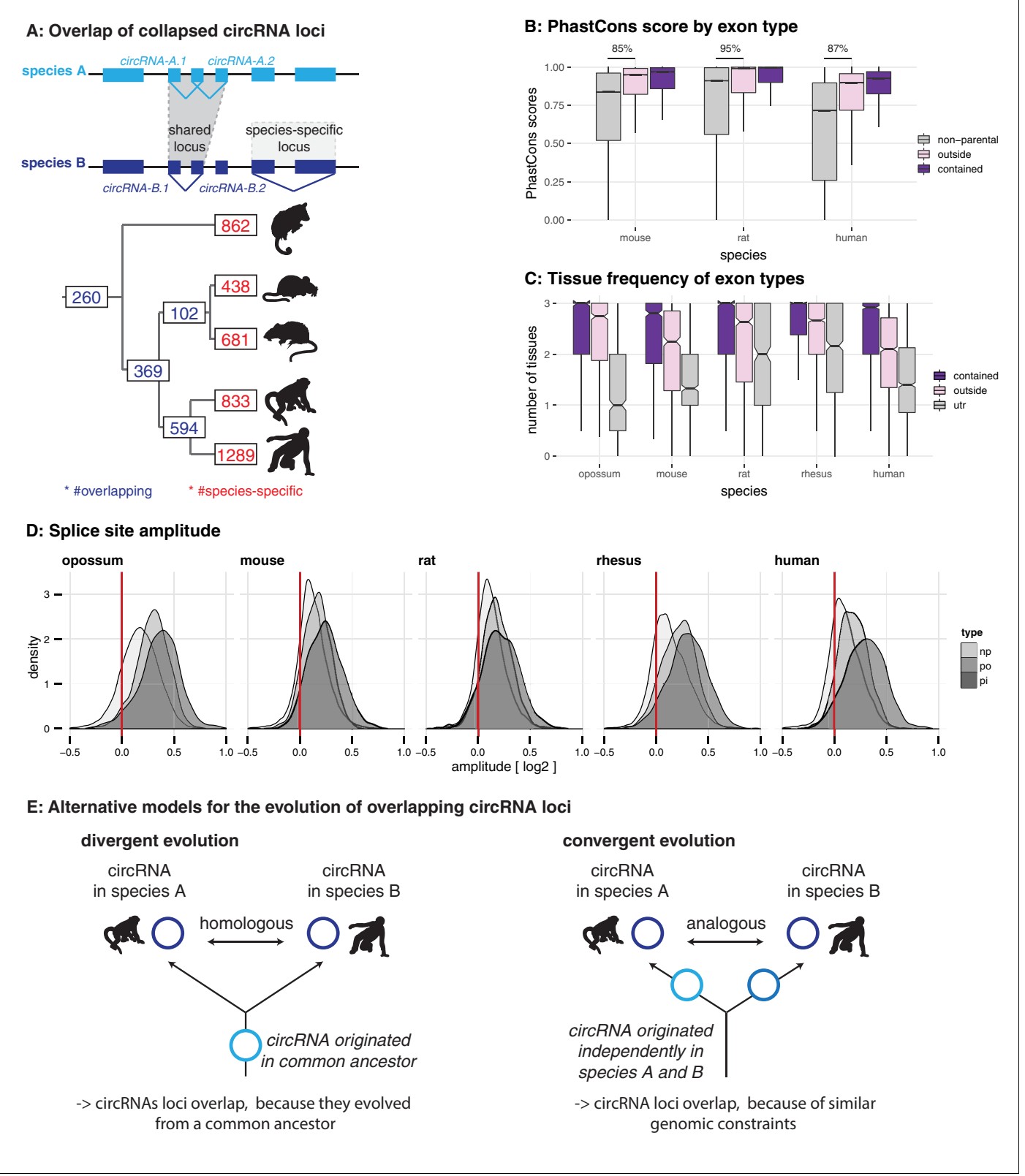

**Figure 2.** Evolutionary properties of circRNAs. (**A**) CircRNA loci overlap between species. Upper panel: Schematic representation of the orthology definition used in our study. CircRNAs were collapsed for each gene, and coordinates were lifted across species. Lower panel: Number of circRNA loci that are species-specific (red) or circRNAs that arise from orthologous exonic loci of 1:1 orthologous genes (i.e. circRNAs sharing 1:1 orthologous exons) across lineages (purple) are counted. We note that in the literature, other circRNA 'orthology' definitions can be found, too. For example, assigning

*Figure 2 continued on next page*

*Figure 2 continued*

circRNA orthology simply based on parental gene orthology implies calling also those circRNAs 'orthologous' that do not share any orthologous exons, which directly argues against the notion of circRNA homology; that is, a common evolutionary origin (see *Figure 2—figure supplement 1A*). Overall, the orthology considerations we applied largely follow the ideas sketched out in *Patop et al., 2019*. (B) Distribution of phastCons scores for different exon types. PhastCons scores were calculated for each exon using the conservation files provided by ensembl. PhastCons scores for non-parental exons (grey), exons in parental genes, but outside of the circRNA (pink) and circRNA exons (purple) are plotted. The difference between circRNA exons and non-parental exons that can be explained by parental non-circRNA exons is indicated above the plot. (C) Mean tissue frequency of different exon types in parental genes. The frequency of UTR exons (grey), non-UTR exons outside of the circRNA (pink) and circRNA exons (purple) that occur in one, two, or three tissues was calculated for each parental gene. (D) Distribution of splice site amplitudes for different exon types. Distribution of median splice site GC amplitude (log2-transformed) is plotted for different exon types (np = non-parental, po = parental, but outside of circRNA, pi = parental and inside circRNA). Red vertical bars indicate values at which exon and intron GC content would be equal. (E) Different evolutionary models explaining the origins of overlapping circRNA loci.

The online version of this article includes the following figure supplement(s) for figure 2:

**Figure supplement 1.** CircRNA loci overlap between species.
**Figure supplement 2.** Amplitude correlations.

human: 61.6%; calculated from hotspot counts in *Figure 1B* and loci counts in *Figure 2A*). Thus, despite applying circRNA enrichment strategies for library preparation and lenient thresholds for computational identification, the number of potentially conserved orthologous circRNAs is surprisingly low. At first sight, this outcome is at odds with previous reports of higher circRNA conservation that were, however, frequently based on more restricted cross-species datasets (e.g. comparison human-mouse in *Rybak-Wolf et al., 2015*). Further analyses confirmed that also in our datasets, it was the use of additional evolutionary species that drove the strong reduction in potentially conserved circRNA candidates – see for example how the addition of the rat or of rhesus macaque datasets affect the human-mouse comparison (*Figure 2—figure supplement 1B*).

We next analysed the properties of circRNA exons and started with phastCons scores, which are based on multiple alignments and known phylogenies and describe conservation levels at single-nucleotide resolution (*Siepel et al., 2005*). To assess whether circRNA exons were distinct from non-circRNA exons in their conservation levels, we calculated phastCons scores for different exon types (circRNA exons, non-circRNA exons, UTR exons). CircRNA exons showed higher phastCons scores than exons from the same genes that were not spliced into circRNAs (*Figure 2B*). This would be the expected outcome if purifying selection acted on functionally conserved circRNAs. However, other mechanisms may be relevant as well; constitutive exons, for example, generally exhibit higher conservation scores than alternative exons (*Modrek and Lee, 2003*; *Ermakova et al., 2006*). We thus analysed exon features in more detail. First, the comparison of phastCons scores between exons of non-parental genes, parental genes and circRNAs revealed that parental genes were per se highly conserved (*Figure 2B*): 85–95% of the observed median differences between circRNA exons and non-parental genes could be explained by the parental gene itself. Next, we compared the usage of parental gene exons across organs (*Figure 2C*). We observed that circRNA exons are more frequently used in isoforms expressed in multiple organs than non-circRNA parental gene exons. Finally, we analysed the sequence composition at the splice sites, which revealed that GC amplitudes (i.e. the differences in GC content at the intron-exon boundary) are significantly higher for circRNA-internal exons than for parental gene exons that were located outside of circRNAs (*Figure 2D*).

Collectively, these observations (i.e. increased phastCons scores, expression in multiple tissues, increased GC amplitudes) prompt the question whether the exon properties associated with circRNAs actually reflect at their core an enrichment for constitutive exons. Under this scenario, the supposed high conservation of circRNAs may not be directly associated with the circRNAs themselves, but with constitutive exons that the circRNAs contain. Thus, even many of the circRNAs 'shared' across species might actually not be homologous. That is, rather than reflecting (divergent) evolution from common ancestral circRNAs (*Figure 2E*, left panel), they may frequently have emerged independently (convergently) during evolution in the lineages leading to the different species, thus potentially representing 'analogous' transcriptional traits (*Figure 2E*, right panel).

## CircRNA parental genes are associated with low GC content and high sequence repetitiveness

To explore whether convergent evolution played a role in the origination of circRNAs, we set out to identify possible structural and/or functional characteristics that may establish a specific genomic environment (a 'parental gene niche') that would potentially favour analogous circRNA production. To this end, we compared GC content and sequence repetitiveness of circRNA parental vs. non-parental genes.

GC content is an important genomic sequence characteristic associated with distinct patterns of gene structure, splicing and function (*Amit et al., 2012*). We realised that the increased GC amplitudes at circRNA exon-intron boundaries (see above, *Figure 2D*) were mainly caused by a local decrease of intronic GC content rather than by an increase in exonic GC content (*Supplementary file 5*, *Figure 2—figure supplement 2*). We subsequently explored the hypothesis that GC content could serve to discriminate parental from non-parental genes and grouped all genes into five categories from low (L) to high (H) GC content (isochores; L1 <37%, L2 37–42%, H1 42–47%, H2 47–52% and H3 >52% GC content) (*Figure 3A*). Non-parental genes displayed a unimodal distribution in the two rodents (peak in H1), were generally GC-poor in opossum (peak in L1), and showed a more complex isochore structure in rhesus macaque and human (peaks in L2 and H3), in agreement with previous findings (*Galtier and Mouchiroud, 1998*; *Mikkelsen et al., 2007*). Notably, circRNA parental genes showed a distinctly different distribution than non-parental genes and a consistent pattern across all five species, with the majority of genes (82–94% depending on species) distributing to the GC-low gene groups, L1 and L2 (*Figure 3A*).

We next analysed intron repetitiveness – a structural feature that has previously been associated with circRNA biogenesis. We used megaBLAST to align all annotated coding genes with themselves in order to identify regions of complementarity in the sense and antisense orientations of the gene (reverse complement sequences, RVCs) (*Ivanov et al., 2015*). We then compared the level of self-complementarity between parental and non-parental genes within the same GC isochore of note, self-complementarity generally shows negative correlations with GC-content. This analysis revealed more pronounced self-complementarity for parental genes than for non-parental genes (*Figure 3B*).

CircRNA parental genes may also show an association with specific functional properties. Using data from three human cell studies (*Steinberg et al., 2015*; *Pai et al., 2012*; *Koren et al., 2012*), our analyses revealed that circRNA parental genes are biased towards early replicating genes, showed higher steady-state expression levels, and are characterised by increased haploinsufficiency scores (*Figure 3—figure supplement 1*). Collectively, we conclude that circRNA parental genes exhibit not only distinct structural features (low GC content, high repetitiveness), but also specific functional properties associated with important roles in human cells.

## Among the multiple predictors of circRNA parental genes, low GC content distinguishes circRNA hotspots

The above analyses established characteristic sequence, conservation and functional features for circRNA parental genes. Using linear regression analyses, we next determined which of these properties represented the main predictor(s). We used parental vs. non-parental gene as the response variable of the model, and several plausible explanatory variables. These were: GC content; exon and transcript counts; genomic length; number of repeat fragments in sense/antisense; expression level; phastCons score; tissue specificity index. After training the model on a data subset (80%), circRNA parental gene predictions were carried out on the remainder of the dataset (20%) (see Materials and methods). Notably, predictions occurred with high precision (accuracy 72–79%, sensitivity of 75%, specificity 71–79% across all species) and uncovered several significantly associated features (*Table 1*, *Supplementary file 6*, *Figure 3—figure supplement 2*). Consistently for all species, the main parental gene predictors are low GC content (log-odds ratio -1.84 to -0.72) and increased number of exons in the gene (log-odds ratio 0.30 to 0.45). Furthermore, features positively associated with circRNA production are increased genomic length (log-odds ratio 0.17 to 0.26), increased proportion of reverse-complementary areas (repeat fragments) within the gene (log-odds ratio 0.20 to 0.59), increased expression levels (log-odds ratio 0.25 to 0.38) and higher phastCons scores (log-odds ratio 0.45 to 0.58) (*Table 1*, *Figure 3C–D*, *Supplementary file 6*). Notably, parental genes of previously reported functional human circRNAs – for example, circHipk3 (*Zheng et al.,*

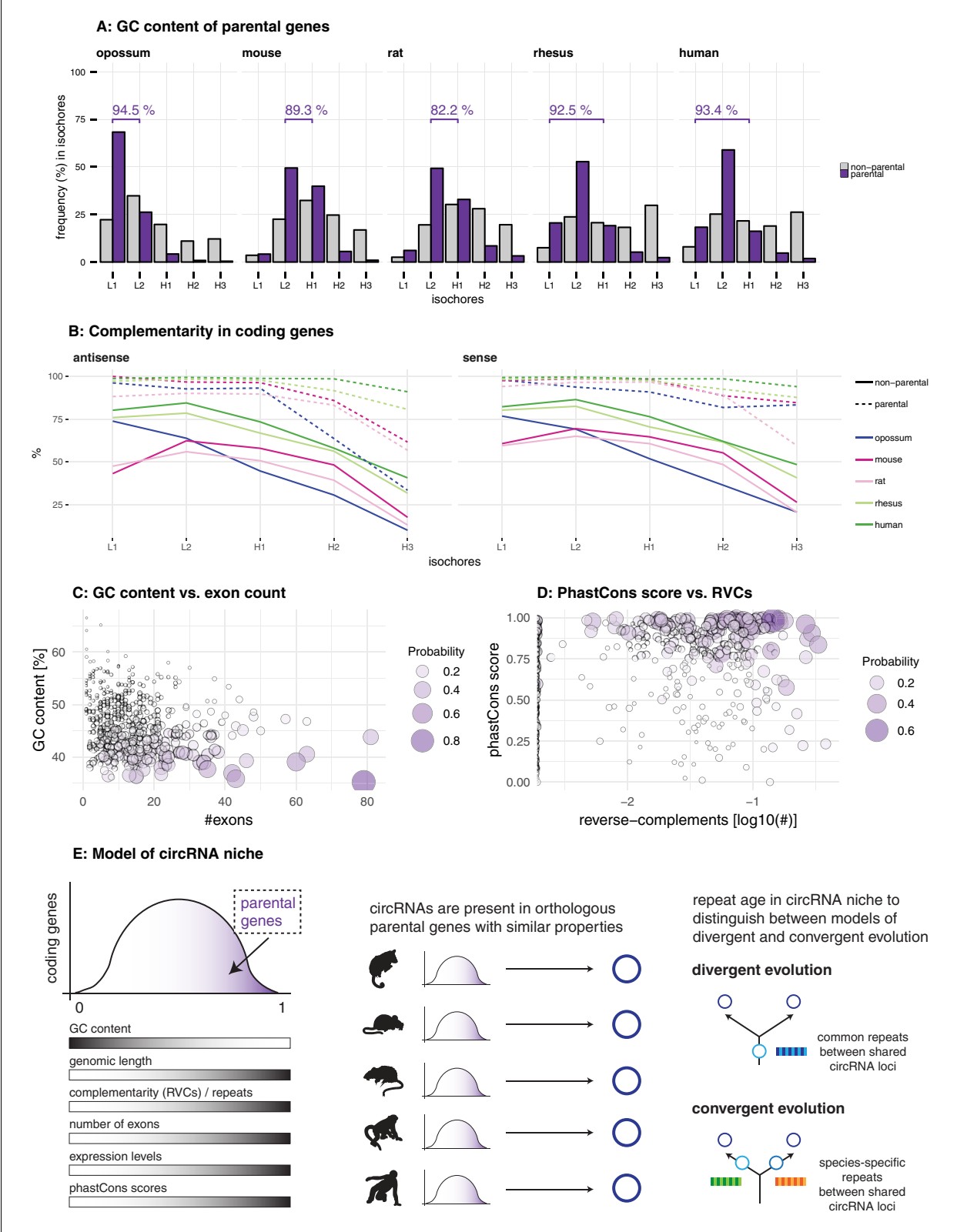

**Figure 3.** Characterisation of circRNA parental gene properties. (**A**) GC content of parental genes. Coding genes were classified into L1-H3 based on their GC content, separately for non-parental (grey) and parental genes (purple). The percentage of parental genes in L1-L2 (opossum, mouse, rat) and L1-H1 (rhesus macaque, human) is indicated above the respective graphs. (**B**) Complementarity in coding genes. Each coding gene was aligned to itself in sense and antisense orientation using megaBLAST. The proportion of each gene involved in an alignment was calculated and plotted against its

*Figure 3 continued on next page*

*Figure 3 continued*

isochore. (C-D) Examples of parental gene predictors for linear regression models. A generalised linear model (GLM) was fitted to predict the probability of the murine coding gene to be parental, whereby x- and y-axis represent the strongest predictors. Colour and size of the discs correspond to the p-values obtained for 500 genes randomly chosen from all mouse coding genes used in the GLM. (E) Model of circRNA niche.

The online version of this article includes the following figure supplement(s) for figure 3:

**Figure supplement 1.** Replication time, gene expression steady-state levels and GHIS of human parental genes.
**Figure supplement 2.** Distribution of prediction values for non-parental and parental circRNA genes.
**Figure supplement 3.** Properties of 'functional circRNAs' from literature.
**Figure supplement 4.** Validation of parental gene GLM on Werfel et al. dataset.
**Figure supplement 5.** Properties of highly expressed circRNAs.

*2016*) and circMbnl1 (*Ashwal-Fluss et al., 2014*) that sequester miRNAs and proteins, respectively – obtain high prediction values in our model and share the above specific properties (*Figure 3—figure*

**Table 1.** Generalised linear model predicting the probability of coding genes to be a parental gene. A generalised linear model was fitted to predict the probability of coding genes to be a parental gene ($n_{opossum}$ = 18807, $n_{mouse}$ = 22015, $n_{rat}$ = 11654, $n_{rhesus}$ = 21891, $n_{human}$ = 21744). The model was trained on 80% of the data (scaled values, cross-validation, 1000 repetitions). Only the best predictors were kept and then used to predict probabilities for the remaining 20% of data points (validation set, shown in table). Genomic length, number of exons and GC content are based on the respective ensembl annotations; number of repeats in antisense and sense orientation to the gene was estimated using the RepeatMasker annotation, phastCons scores taken from UCSC (not available for opossum and rhesus macaque) and expression levels and the tissue specificity index based on *Brawand et al., 2011*. An overview of all log-odds ratios and p-values calculated in the validation set of each species is provided in the table, further details can be found in *Supplementary file 6*. Abbreviations: md = opossum, mm = mouse, rn = rat, rm = rhesus macaque, hs = human. Significance levels: '***' < 0.001, '**' < 0.01, '*' < 0.05, 'ns' >= 0.05.

| Predictor | Log-odds range (significance) | Species with significant predictor |
|---|---|---|
| Genomic gene length (bp) | rn: 0.26 (***)<br>rm: 0.17 (***)<br>hs: 0.26 (***)<br>md, mm: ns | rn, rm, hs |
| Number of exons | md: 0.45 (***)<br>mm: 0.38 (***)<br>rn: 0.30 (***)<br>rm: 0.42 (***)<br>hs: 0.32 (***) | md, mm, rn, rm, hs |
| GC content | md: -1.84(***)<br>mm: -1.09(***)<br>rn: -0.72(***)<br>rm: -1.44(***)<br>hs: -1.42(***) | md, mm, rn, rm, hs |
| Repeat fragments (antisense) | md: 0.28 (**)<br>mm: 0.20 (**)<br>rm: 0.59 (***)<br>rn, hs: ns | md, mm, rm |
| Repeat fragments (sense) | hs: 0.58 (***)<br>md, mm, rn, rm: ns | hs |
| PhastCons scores | mm: 0.58 (***)<br>rn: 0.51 (***)<br>hs: 0.45 (***) | mm, rn, hs |
| Mean expression levels | md: 0.34 (**)<br>rm: 0.38 (***)<br>hs: 0.25 (**)<br>mm, rn: ns | md, rm, hs |
| Tissue specificity index | md, mm, rn, rm, hs: ns | - |

*supplement 3*). In addition, the identified circRNA parental gene predictors were not restricted to our datasets but could be determined from independent circRNA data as well. Thus, the analysis of mouse and human heart tissue data (*Werfel et al., 2016*) – on which our linear regression models predicted parental genes with comparable accuracy (74%), sensitivity (75%), and specificity (74%) – revealed that circRNA parental genes were low in GC content, exon-rich, and showed enrichment for repeats (*Figure 3—figure supplement 4*). In conclusion, the identified properties likely represent generic characteristics of circRNA parental genes that are suitable to distinguish them from non-parental genes.

Many circRNAs are formed from circRNA hotspots (*Figure 1C*). We therefore asked whether among the features that our regression analysis identified for parental genes, some would be suitable to further distinguish hotspots. First, we assessed whether hotspots were more likely to be shared between species than parental genes that produced only a single circRNA isoform. The applied regression model indeed detected a positive correlation between the probability of a parental gene being a hotspot and having orthologous parental genes across multiple species (*Supplementary file 7*); moreover, log-odds ratios increased with the distance and number of species across which the hotspot was shared (e.g. mouse: 0.29 for shared within rodents, 0.67 for shared with eutherian species and 0.72 for shared within therian species). We next interrogated whether any particular feature would be able to specify circRNA hotspots among parental genes. A single factor, low GC content, emerged as a consistent predictor for circRNA hotspots among all circRNA-generating loci (*Supplementary file 8*). As expected, the predictive power was lower than that of the previous models, which were designed to discriminate parental vs. non-parental genes and which had identified low GC content as well. These findings imply that hotspots emerge across species in orthologous loci that offer similarly favourable conditions for circRNA formation, most importantly low GC content. The increased number of circRNAs that become detectable when CPM thresholds are lowered (see above, *Figure 1C*) is also in agreement with the sporadic formation of different circRNAs whenever genomic circumstances allow for it. Overall, our observations suggest that differences between hotspot and non-hotspot loci, or between high and low abundance circRNAs, are quantitative rather than qualitative in nature. Thus, the comparison of high vs. low expression circRNAs (based on 90% expression quantile; below = low, above = high expression) indicated the same set of properties, albeit amplified, in the highly expressed circRNAs (*Supplementary file 9*). Parental genes of highly expressed circRNAs in opossum, rhesus macaque and human yielded higher prediction values in our generalised linear model, which was consistently driven by low GC content (*Supplementary file 9*). High expression circRNAs were also more likely to be expressed in all three tissues (*Figure 3—figure supplement 5A*) and to originate from a hotspot (*Figure 3—figure supplement 5B*), and they were more often shared across multiple species (*Figure 3—figure supplement 5C*, *Supplementary file 10*).

Collectively, our analyses thus reveal that circRNA parental genes are characterised by a set of distinct features: low GC content, increased genomic length and number of exons, higher expression levels and increased phastCons scores (*Figure 3E*). These features were detected independently across species, suggesting the presence of a unique, syntenic genomic niche in which circRNAs can be produced ('circRNA niche'). While helpful to understand the genomic context of circRNA production, these findings do not yet allow us to distinguish between the two alternative models of divergent and convergent circRNA evolution (*Figure 2E*). To elucidate the evolutionary trajectory and timeline underlying the emergence of the circRNAs, we sought to scrutinize the identified feature 'complementarity and repetitiveness' of the circRNA niche. Previous studies have associated repetitiveness with an over-representation of small TEs – such as primate Alu elements or the murine B1 elements – in circRNA-flanking introns; these TEs may facilitate circRNA formation by providing RVCs that are the basis for intramolecular base-pairing of nascent RNA molecules (*Ivanov et al., 2015*; *Jeck et al., 2013*; *Zhang et al., 2014*; *Wilusz, 2015*; *Liang and Wilusz, 2014*). Interestingly, while the biogenesis of human circRNAs has so far been mainly associated with the primate-specific (i.e. evolutionarily young) Alu elements, a recent study has highlighted several circRNAs that rely on the presence of the more ancient, mammalian MIR elements (*Yoshimoto et al., 2020*). A comprehensive understanding of the evolutionary age of TEs in circRNA-flanking introns could thus provide important insights into the modes of circRNA emergence: the presence of common (i.e. old) repeats would point towards divergent evolution of circRNAs from a common circRNA ancestor, whereas an

over-representation of species-specific (i.e. recent) repeats would support the notion of convergent circRNA evolution (*Figure 3E*).

## CircRNA flanking introns are enriched in species-specific TEs

Using our cross-species datasets, we investigated the properties and composition of the repeat landscape relevant for circRNA biogenesis – features that have remained poorly characterised so far. As a first step, we generated for each species a background set of 'control introns' from non-circRNA genes that were matched to the circRNA flanking introns in terms of length distribution and GC content. We then compared the abundance of different repeat families within the two intron groups. In all species, TEs belonging to the class of Short Interspersed Nuclear Elements (SINEs) are enriched within the circRNA flanking introns as compared to the control introns. Remarkably, the resulting TE enrichment profiles were exquisitely lineage-specific, and even largely species-specific (*Figure 4A*). In mouse, for instance, the order of enrichment is from the B1 class of rodent-specific B elements (strongest enrichment and highest frequency of >7.5 TEs per flanking intron) to B2 and B4 SINEs. In rat, B1 (strong enrichment, yet less frequent than in mouse) is followed by ID (Identifier) elements, which are a family of small TEs characterised by a recent, strong amplification history in the rat lineage (*Kim et al., 1994*; *Kim and Deininger, 1996*); B2 and B4 SINEs only followed in 3rd and 4th position. In rhesus macaque and human, Alu elements are the most frequent and strongly enriched TEs (around 14 TEs per intron), consistent with the known strong amplification history in the common primate ancestor (reviewed in *Batzer and Deininger, 2002*; *Figure 4A*). The opossum genome is known for its high number of TEs, many of which may have undergone a very species-specific amplification pattern (*Mikkelsen et al., 2007*). This is reflected in the distinct opossum enrichment profile (*Figure 4—figure supplement 1*).

As pointed out above, TEs are relevant for circRNA formation because they can provide RVCs for the intramolecular base-pairing of nascent RNA molecules (*Ivanov et al., 2015*; *Jeck et al., 2013*; *Zhang et al., 2014*; *Wilusz, 2015*; *Liang and Wilusz, 2014*). Pre-mRNA folding into a hairpin with a paired stem (formed by the flanking introns via the dimerised RVCs) and an unpaired loop region (carrying the future circRNA) leads to a configuration that brings backsplice donor and acceptor sites into close proximity, thus facilitating circRNA formation. In order to serve as efficient RVCs via this mechanism, TEs likely need to fulfil certain criteria. Thus, the dimerisation potential is expected to depend on TE identity, frequency, and position. In the simplest case, two integration events involving the same TE (in reverse orientation) will lead to an extended RVC stretch. Yet also different transposons belonging to the same TE family will show a certain degree of sequence similarity that depends on their phylogenetic distance; sequence differences that have evolved are likely to compromise the base-pairing potential. To account for such effects, we sought to calculate the actual binding energies for RVC interactions and combine this analysis with phylogenetic distance information, thus potentially allowing us to detect the most likely drivers of circRNA formation, as well as their evolutionary age.

Our analyses revealed that relatively few specific dimers represented the majority of all predicted dimers (i.e. top-5 dimers accounted for 78% of all dimers in flanking introns in opossum, and for 50%, 55%, 43%, and 38% in mouse, rat, rhesus macaque and human, respectively) (*Figure 4B*). Given the high abundance of young, still active transposons in the respective genomes (*Figure 4A*), we suspected that simply basing our further analyses of dimerisation potential on phylogenetic distance between different TEs would not provide sufficient resolution. Indeed, as shown for mouse (*Figure 4C–D*), phylogenetic age separates large subgroups, but not TEs of the same family whose sequences have diverged by relatively few nucleotides. By contrast, classification by binding affinities creates more precise, smaller subgroups that lack, however, the information on phylogenetic age (*Figure 4E*). Therefore, we combined both age and binding affinity information into an overall 'pairing score' (see Materials and methods). Principal component analysis (PCA) showed that this measure efficiently separated different TE families and individual family members, with PC1 and PC2 explaining approximately 76% of observed variance (*Figure 4F*; *Figure 4—figure supplement 2*). Importantly, this analysis suggests that the most frequently occurring dimers (top-5 dimers are depicted with blue connecting lines in *Figure 4F*) are formed by recently active TE family members. In mouse, an illustrative example are the dimers formed by the B1_Mm, B1_Mus1, and B1_Mus2 elements (*Figure 4F*), which are among the most recent (and still active) TEs in this species (*Figure 4C*). Across species, our analyses allowed for the same conclusions. For example, the dominant dimers in

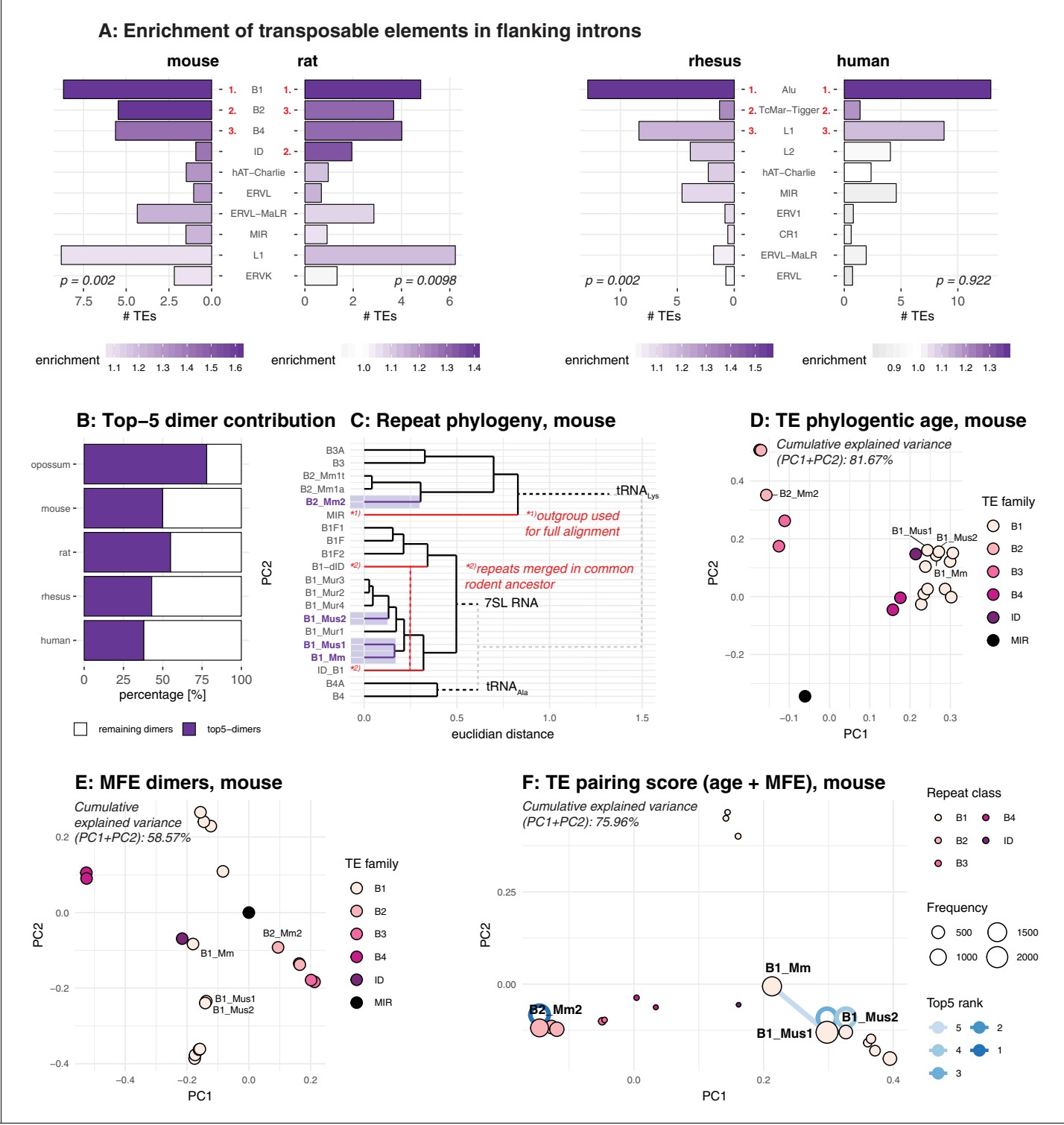

**Figure 4.** Analysis of the repeat landscape of circRNA parental genes. (**A**) Enrichment of TEs in flanking introns for mouse, rat, rhesus macaque and human. The number of TEs was quantified in both intron groups (circRNA flanking introns and length- and GC-matched control introns). Enrichment of TEs is represented by colour from high (dark purple) to low (grey). The red numbers next to the TE name indicate the top-3 enriched TEs in each species. Enrichment was assessed using a Wilcoxon Signed Rank Test; p-values are indicated at the bottom of each plot. (**B**) Top-5 dimer contribution. The graph shows the proportion of top-5 dimers (purple) vs. other, remaining dimers (white) to all predicted dimers in flanking introns. Top-5 dimers thus account for 78, 50, 55, 43, and 38% of all dimers in opossum, mouse, rat, rhesus macaque and human, respectively. (**C**) Phylogeny of mouse TEs. Clustal-alignment based on consensus sequences of TEs. Most recent TEs are highlighted. (**D**) PCA for phylogenetic age of mouse TE families. PCA is

*Figure 4 continued on next page*

*Figure 4 continued*

based on the clustal-alignment distance matrix for the reference sequences of all major SINE families in mouse with the MIR family used as an outgroup. TEs present in the top-5 dimers are labelled. (E) PCA based on binding affinity of mouse TE families. PCA is based on the minimal free energy (MFE) for all major SINE families in mouse with the MIR family used as an outgroup. TEs present in the top-5 dimers are labelled. (F) PCA for TE pairing score of mouse dimers. PCA is based on a merged and normalised score, taking into account binding strength of the dimer structure (=MFE) and phylogenetic distance. Absolute frequency of TEs is visualised by circle size. TEs present in the five most frequent dimers (top-5) are highlighted by blue lines connecting the two TEs engaged in a dimer (most frequent dimer in dark blue = rank 1). If the dimer is composed of the same TE family members, the blue line loops back to the TE (=blue circle).

The online version of this article includes the following figure supplement(s) for figure 4:

**Figure supplement 1.** Enrichment of transposable elements in flanking introns for opossum.

**Figure supplement 2.** PCA and phylogeny of opossum, rat, rhesus macaque, and human repeat dimers.

rat were the recently amplified ID elements, and not the more abundant (yet older in their amplification history) B1 family of TEs (*Figure 4—figure supplement 2B*; *Kim et al., 1994*; *Kim and Deininger, 1996*). In opossum, the most prominent dimers consisted of opossum-specific SINE1 elements, which are similar to the Alu elements in primates, but possess an independent origin (*Figure 4—figure supplement 2A*; *Gu et al., 2007*). Finally, within the primate lineage, the dimer composition was more uniform, probably due to the high amplification rate of the AluS subfamily (>650000 copies) in the common ancestor of Old World monkeys and the relatively recent divergence time of macaque and human (*Figure 4—figure supplement 2C–D*; *Deininger, 2011*).

In conclusion, the above analyses of RVCs revealed that dimer-forming sequences in circRNA flanking introns were most frequently composed of recent, and often currently still active, TEs. Therefore, the dimer repertoires were specific to the lineages (marsupials, rodents, primates) and/or (as most clearly visible within the rodent lineage) even species-specific.

## Flanking introns of shared circRNA loci are enriched in evolutionarily young TEs

We next compared the dimer composition of introns from shared vs. species-specific circRNA loci. We reasoned that in the case of shared circRNA loci that have evolved from a common, ancestral circRNA, we would detect evidence for evolutionarily older TE integration events and shared dimers as compared to species-specific, younger circRNA loci. For our analysis, we took into account the frequency, enrichment, and age of the TEs and, moreover, their degradation rate (milliDiv; see below) and the minimal free energy (MFE) of the dimer structure.

First, we analysed the dimer composition of flanking introns in shared and species-specific circRNA loci. We extracted the top-100 most and least frequent dimers of all circRNA loci, and compared their enrichment factors and mean age (categorised for simplicity into four groups: 1 = species-specific, 2 = lineage-specific, 3 = eutherian, 4 = therian) across the two groups of parental genes (shared and species-specific). The analysis revealed that the most frequent dimers are consistently formed by the youngest elements in both groups of genes, and that the frequency distribution of the top-100 dimers was significantly different between species (see *Figure 5A* for rat and human; other species in *Figure 5—figure supplement 1*). In rat, for instance, all top-5 dimers are composed of repeats from the youngest ID family members; in human, dimers involving AluY elements are strongly enriched (*Figure 5A*). On average, most dimers occur at least once or twice per shared circRNA gene, corresponding to a 1.4- to 2.1-fold enrichment in comparison to species-specific circRNA loci (*Supplementary file 11*). Conceivably, the multiple resulting dimerisation possibilities could act cumulatively to position circRNA exons for backsplicing. Furthermore, we observed that many RVCs overlapped each other, so that one repeat in one RVC could dimerise with different repeats in multiple other RVCs. Due to the increased frequency of young repeat elements in shared circRNA loci, these 'co-pairing possibilities' further increase the number of possible dimers that can be formed (*Figure 5—figure supplement 2*). A representative example for a shared circRNA-generating locus with its complex dimer interaction landscape, involving young species-specific repeats, is the *Akt3* locus (*Figure 5B*). Thus, although *Akt3* circRNAs are shared between human (upper panel), mouse (middle panel), and opossum (lower panel), the dimer landscapes are entirely specifies-specific (see top-5 dimers that are highlighted in the figure).

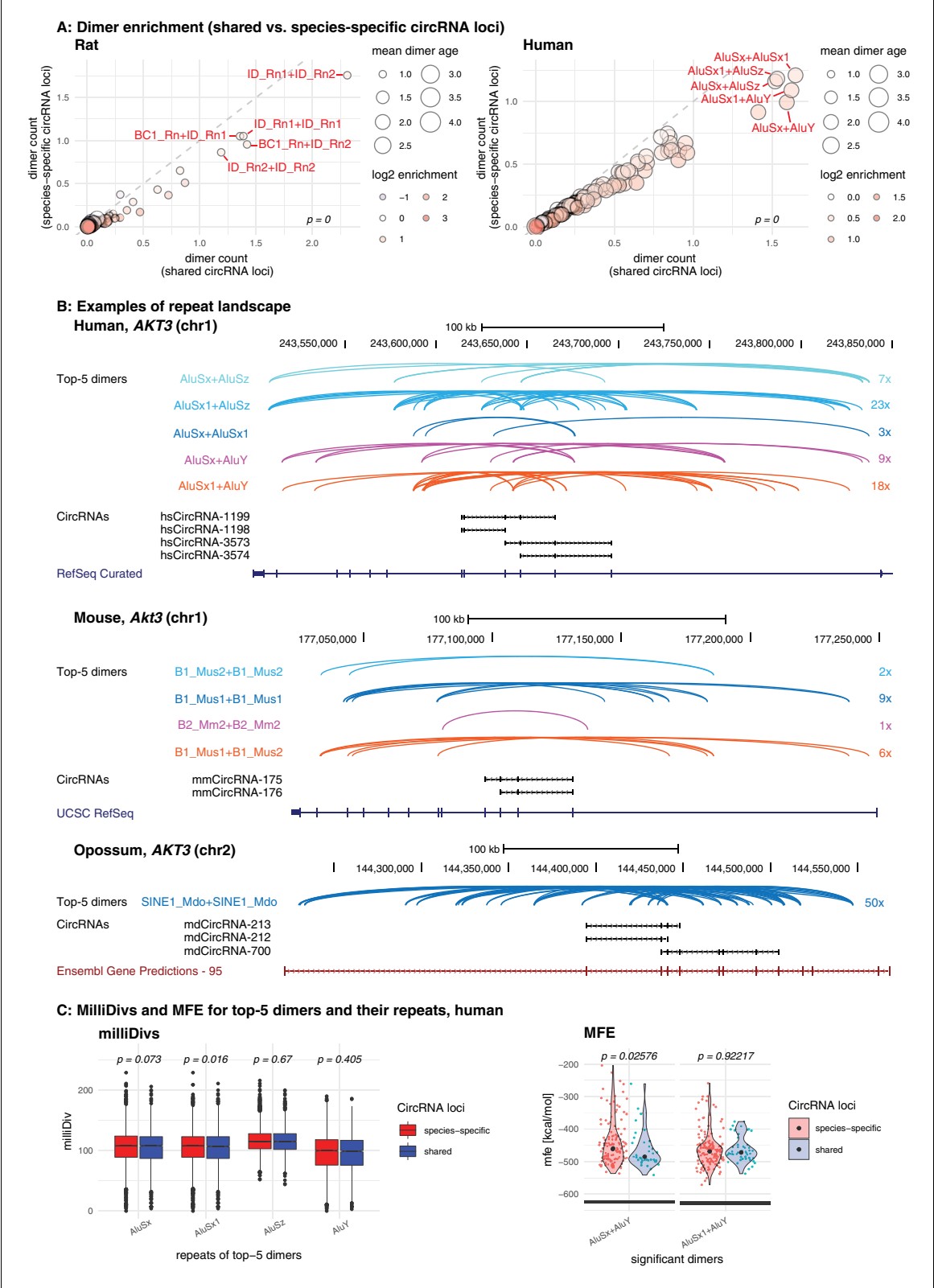

**Figure 5.** Repeat analysis and dimer potential of shared and species-specific parental genes. (**A**) Dimer enrichment in shared vs. species-specific repeats in rat and human (see *Figure 5—figure supplement 1* for other species). The frequency (number of detected dimers in a given parental gene), log2-enrichment (shared vs. species-specific) and mean age (defined as whether repeats are species-specific: age = 1, lineage-specific: age = 2, eutherian: age = 3, therian: age = 4) of the top-100 most frequent and least frequent dimers in parental genes with shared and species-specific circRNA

*Figure 5 continued on next page*

*Figure 5 continued*

loci in rat and human were analysed. The frequency is plotted on the x- and y-axis, point size reflects the age and point colour the enrichment (blue = decrease, red = increase). Based on the comparison between shared and species-specific dimers (using a Wilcoxon Signed Rank Test), the top-5 dimers defined by frequency and enrichment are highlighted and labelled in red. (B) Species-specific dimer landscape for the *Akt3* gene in human, mouse and opossum. UCSC genome browser view for the parental gene, circRNAs and top-5 dimers (as defined in panel B). Start and stop positions of each dimer are connected via an arc. Dimers are grouped by composition represented by different colours, the number of collapsed dimers is indicated to the right-side of the dimer group. Only dimers that start before and stop after a circRNAs are shown as these are potentially those that can contribute to the hairpin structure. The human *Akt3* gene possesses two circRNA clusters. For better visualisation, only the upstream cluster is shown. (C) Degradation rates (MilliDivs) and minimal free energy (MFE) for top-5 dimers in human. MilliDiv values for all repeats composing the top-5 dimers (defined by their presence in all parental genes) were compared between parental genes of species-specific (red) and shared (blue) circRNA loci in human (see *Figure 5—figure supplement 3* for other species). A Wilcoxon Signed Rank Test was used to compare dimers between parental genes with shared and species-specific circRNA loci, with p-values plotted above the boxplots. MFE values were compared between the least degraded dimers in parental genes of species-specific (red) and shared (blue) circRNA loci. MFE values were calculated using the genomic sequences of all top-5 dimers. For each parental gene, the least degraded dimer (based on its mean milliDiv value) was then chosen which let to a strong enrichment of only a subset of the top-5 dimers (in this case AluSx+AluY and AluSx1+AluY). If enough observations for a statistical test were present, the two distributions (shared/species-specific) were compared using a Student's t-Test and plotted as violin plots with p-values above the plot.

The online version of this article includes the following figure supplement(s) for figure 5:

**Figure supplement 1.** Contribution of species-specific repeats to the formation of shared circRNA loci.
**Figure supplement 2.** Repeat interaction landscape in shared vs. species-specific circRNA loci.
**Figure supplement 3.** MilliDivs and MFE for dimers in shared and species-specific circRNA loci.

The above observations suggest that circRNA-producing genes act as 'transposon sinks' that are prone to insertions of active repeats. Continuously attracting new transposons could contribute to the mechanism that sustains backsplicing and underlies reproducible circRNA expression levels. Moreover, through the recurring addition of new functional repeats, new dimerisation potential would be generated that could make older TEs redundant and allow them to rapidly degrade, thus explaining why ancient TE integration events are no longer detectable. If a circRNA is functionally important for the organism, especially the young, dimerisation-competent repeats may evolve under purifying selection and maintain their pairing ability. We therefore reasoned that low degradation rates in young dimers of shared circRNA loci could hint at functionality. We followed up this idea by analysing the degradation rates of repeats based on their milliDiv values. Briefly, the RepeatMasker annotations (*Smit et al., 2013*) (http://repeatmasker.org; see Materials and methods) provide a quantification of how many 'base mismatches in parts per thousand' have occurred between each specific repeat copy in its genomic context and the repeat reference sequence. This deviation from the consensus sequence is expressed as the milliDiv value. Thus, a high milliDiv value implies that a repeat is strongly degraded, typically due to its age (the older the repeat, the more time its sequence has had to diverge). Low milliDiv values suggest that the repeat is younger (i.e. it had less time to accumulate mutations) or that purifying selection prevented the accumulation of mutations.

Following this rationale, we determined in each species the degradation rates for the repeats forming the top-5 dimers. Comparing their milliDiv values species-specific parental genes revealed no significant differences in any of the species (*Figure 5C* – left panel, *Figure 5—figure supplement 3* – left panel). Because degradation rates alone may not fully capture the actual decline in pairing strength within a dimer (e.g. compensatory changes and dimer length are not/poorly accounted for), we further analysed actual binding energies. To this end, we selected the least-degraded dimer for every parental gene in both groups (shared/species-specific) and calculated the minimal free energies (MFEs) of dimer formation. We detected no difference between the groups, suggesting that dimers of shared circRNA loci are not subject to a specific selection pressure, but degrade identically to dimers in species-specific circRNA loci (*Figure 5C* – right panel, *Figure 5—figure supplement 3* – right panel). Furthermore, we observed that dimers comprising 'intermediate age' repeats (i.e. B1_Mur2, B1_Mur3, B1_Mur4, present in Muridae) could be found in the species-specific 'least-degraded' dimers, yet they were absent from the shared group, which rather contained the top-1/top-2 most enriched and youngest dimers (e.g. AluSx+AluY and AluSx1+AluY in human *Figure 5C*; ID_Rn1+ID_Rn1 and ID_Rn1+ID_Rn2 in rat) (*Figure 5C*, *Figure 5—figure supplement 3C*).

Taken together, we conclude that circRNAs are preferentially formed from loci that have attracted transposons in recent evolutionary history. Even in the case of shared circRNA loci the

actual repeat landscapes, dimer predictions, transposon ages and degradation rates, as well as RVC pairing energies, are most consistent with the model that circRNAs are analogous features that have been formed by convergent evolution, rather than homologous features originating from a common circRNA ancestor.

## Discussion

Different mechanistic scenarios to explain the origins and evolution of circRNAs have been considered in the field (reviewed in *Patop et al., 2019*). In our study, we have investigated this topic through the analysis of novel, dedicated cross-species datasets. Notably, we propose that many circRNAs have not evolved from common, ancestral circRNA loci, but have emerged independently through convergent evolution, most likely driven by structural commonalities of their parental genes. Thus, the modelling of parental genes uncovered features that are associated with circRNA biogenesis, in support of the concept of a 'circRNA niche' in which circRNAs are more likely to be generated: genetic loci giving rise to circRNAs are generally long, exon-rich and located in genomic regions of low GC content. In the case of orthologous parental genes, these structural characteristics are shared as well, and they have led to shared integration biases for transposons, that is to shared, genomic 'TE hotspots'.

It is well established that intronic TE insertions are critical for circRNA biogenesis as they provide reverse-complementary sequences for intramolecular pre-mRNA folding via TE dimers, giving rise to the secondary structures that facilitate productive backsplicing. Important new insights that our study provides on circRNA evolution come from the deep analysis of the transposon landscapes, including the TE identities, their ages, degradation rates and dimerisation potentials. Thus, because the actual TEs predicted as most relevant for dimerisation are mostly not shared across species and are evolutionarily young, we propose that the resulting circRNAs are evolutionarily young as well. In line with this interpretation, circRNAs from orthologous genes frequently do not involve exactly the same 5' and 3' backsplice sites and thus do not encompass precisely the same orthologous exons, but show partial exon overlap across species (see *Figure 2—figure supplement 1*). These findings all argue for a model of convergent evolution at shared circRNA loci, with circRNAs and TEs co-evolving in a species-specific and dynamic manner.

Our model provides an explanation for how circRNAs can arise from orthologous exonic loci across species even if they themselves are not homologous (i.e. they do not stem from common evolutionary precursors that emerged in common ancestors). Importantly, if most circRNAs are evolutionarily young, then, by extension, it is overall rather unlikely that they fulfil crucial functions. This idea is in agreement with the generally low expression levels of circRNAs that have been reported and with accumulation patterns that are frequently tissue-specific and confined to post-mitotic cells (*Guo et al., 2014*; *Westholm et al., 2014*). Importantly, these and other main conclusions of our study overlap with those of two independent manuscripts (with complementary data and analyses) that have appeared in press (*Xu and Zhang, 2021*) and as a publication preprint (*Santos-Rodriguez et al., 2021*), respectively, while we were preparing the revised version of our manuscript.

Why is it frequently the same (orthologous) genes that produce circRNAs, and why do the circRNA hotspots often overlap between species, that is they share common exons? A plausible explanation lies in how TE integration is tolerated. Briefly, intronic TE integration in the vicinity of an intron-exon boundary will likely alter local GC content. For example, GC-rich SINE elements integrating close to a splice site would locally increase intronic GC and thereby decrease the GC amplitude at the intron-exon boundary. Especially in GC-low environments, this can interfere with the intron-defined mechanism of splicing and cause mis-splicing (*Amit et al., 2012*). By contrast, TE integration close to a very strong splice site with a strong GC amplitude – as typically found in canonical exons – would have lower impact. Hence, it would be tolerated better than integration close to alternative exons, whose GC amplitudes are less pronounced. Indeed, our analyses show that circRNA exons are typically canonical exons with strong GC amplitudes. While at first sight, circRNA exons thus appear to be endowed with rather specific, evolutionarily relevant properties – most notably with increased phastCons scores – it is probable that these are a mere consequence of a higher tolerance for TE integration in introns flanking canonical exons.

Many additional characteristics associated with circRNAs – identified in this study or previously by others – can be linked to how the impact of TEs on splicing and transcript integrity is likely to be tolerated. Depending on the site of TE integration, potentially hazardous 'transcript noise' will arise, and these instances will be subject to purifying selection. In particular, TE integration into exons (changing the coding sequence) or directly into splice sites (affecting splicing patterns) will lead to erroneous transcripts (*Zhang et al., 2011*). Thus, the probability that an integration event is tolerated, will be overall lower in short and compact genes as compared to genes with long introns; of note, long genes are also GC-poor (*Zhu et al., 2009*). These characteristics overlap precisely with those that we identify for circRNAs, which are also frequently generated from GC-poor genes with long introns, complex gene structures, and that contain many TEs.

An interesting feature – not analysed in our study, but previously associated with circRNAs – is RNA editing. In particular, introns bracketing circRNAs are enriched in A-to-I RNA editing events, and the RNA-editing enzyme ADAR1 has been reported as a specific regulator of circRNA expression (*Ivanov et al., 2015*; *Rybak-Wolf et al., 2015*). However, A-to-I editing is also a well-known defense mechanism that has evolved to suppress TE amplification. For example, A-to-I RNA editing is associated with intronic Alu elements to inhibit Alu dimers (*Lev-Maor et al., 2008*; *Athanasiadis et al., 2004*). Therefore, it is quite likely that associations between RNA editing and circRNA abundances are a secondary effect from the primary purpose of A-to-I editing, namely the inhibition of Alu amplification. A similar case can be made for DNA methylation that interferes with TE amplification (*Yoder et al., 1997*) and has been linked to circRNA production (*Enuka et al., 2016*). Or, in the case of $N^6$-methyladenosine (m$^6$A), it has recently been proposed that this highly prevalent RNA modification is also involved in dynamically regulating circRNA abundances (*Zhou et al., 2017*; *Park et al., 2019*; *Di Timoteo et al., 2020*). Yet the link of circRNAs to m$^6$A, which is known to influence many steps of mRNA metabolism (reviewed in *Zaccara et al., 2019*; *Lee et al., 2020*), may simply reflect the general targeting of erroneous transcripts for degradation.

In summary, our evolutionary data and the above considerations lead us to conclude that many circRNAs are likely a form of transcript noise – or, more precisely, of mis-splicing – that is provoked by TE integration into parental genes. This conclusion is in full agreement with the observation that in rat neurons, there is a direct correspondence between the pharmacological inhibition of canonical splicing and increased circRNA formation, preferentially affecting circRNAs with long introns and many transposons/RVCs (*Wang et al., 2019*). Altogether, these conclusions make it likely that the majority of circRNAs do not have specific molecular functions, although functional circRNAs have arisen during evolution, as demonstrated in several studies (e.g. *Hansen et al., 2013*; *Conn et al., 2015*; *Du et al., 2016*), presumably from initially non-functional (noise) variants whose emergence was facilitated by the aforementioned mechanisms. During this process, a functional circRNA may ultimately even become independent from the original RVC-based regulation. Evolving from a sequence-based backsplice mechanism to a protein-based one (i.e. relying on RNA-binding proteins, RBPs) could render regulation more versatile and more controllable. Indeed, RBPs have emerged as important regulators of several circRNAs (see e.g. *Ashwal-Fluss et al., 2014*; *Conn et al., 2015*; *Okholm et al., 2020*). The functions of circRNAs seem to be diverse and may often involve the positive or negative regulation of their own parental genes at different expression layers (transcription/ splicing, translation, post-translational modification) through various mechanisms (e.g. competition with linear mRNA splicing, microRNA sponge effects, mRNA traps) (*Shao et al., 2021*). For several of these functional roles, the exact exons/exon portions that form the circRNA, or which elements in the flanking introns drive the process, may not be important, but rather the general maintenance of circularization at a locus during evolution. In this way, diverting mRNA output to non-functional, dead-end circular transcripts could for example represent a mechanism to limit parental gene expression or to control genes that have transformed into transposon sinks.

Finally, we would like to note that circRNAs have emerged as reliable disease biomarkers (*Memczak et al., 2015*; *Bahn et al., 2015*), and their utility for such predictive purposes is not diminished by our conclusion that most circRNAs are unlikely to fulfil direct functions – on the contrary. Even if an altered circRNA profile will likely not indicate causal involvement in a disease, it could hint at misregulated transcription or splicing of the parental gene, at a novel TE integration event, or at problems with RNA editing or methylation machineries. The careful analysis of the circRNA landscape may thus teach us about factors contributing to diseases in a causal fashion even if many or perhaps most circRNAs may not be functional but rather represent transcript noise.

## Materials and methods

### Data deposition, programmes, and working environment

The raw data and processed data files discussed in this publication have been deposited in NCBI's Gene Expression Omnibus (*Edgar et al., 2002*) and are accessible through the GEO Series accession number GSE162152. All scripts used to produce the main figures and tables of this publication have been deposited in the Git Repository circRNA_paperScripts (https://github.com/Frenzchen/circRNA_paperScripts; *Gruhl, 2021*, copy archived at swh:1:rev:51584e2a107500b1a5807218a6-ba4cc811d108f6). This Git repository also holds information on how to run the scripts, and links to the underlying data files for the main figures. The custom pipeline developed for the circRNA identification can be found in the Git Repository ncSplice_circRNAdetection (https://github.com/Frenzchen/ncSplice_circRNAdetection; *Gruhl, 2017*). External programmes used for analyses are listed in *Table 2*.

### Library preparation and sequencing

We used 5 µg of RNA per sample as starting material for all libraries. For each biological replicate (=*tissue X* of *Animal 1* of a given species) two samples were taken: sample one was left untreated, sample two was treated with 20 U RNase R (Epicentre/Illumina, Cat. No. RNR07250) for 1 hr at 37°C to degrade linear RNAs, followed by RNA purification with the RNA Clean and Concentrator-5 kit (Zymo Research) according to the manufacturer's protocol. Paired-end sequencing libraries were prepared from the purified RNA with the Illumina TruSeq Stranded Total RNA kit with Ribo-Zero Gold according to the protocol with the following modifications to select larger fragments: (1) Instead of the recommended 8 min at 68°C for fragmentation, we incubated samples for only 4 min at 68°C to increase the fragment size; (2) In the final PCR clean-up after enrichment of the DNA fragments, we changed the 1:1 ratio of DNA to AMPure XP Beads to a 0.7:1 ratio to select for binding of larger fragments. Libraries were analysed on the fragment analyzer for their quality and sequenced with the Illumina HiSeq 2500 platform (multiplexed, 100 cycles, paired-end, read length 100 nt).

### Identification and quantification of circRNAs

#### Mapping of RNA-seq data

The ensembl annotations for opossum (monDom5), mouse (mm10), rat (rn5), rhesus macaque (rheMac2) and human (hg38) were downloaded from Ensembl (see *Table 3*) to build transcriptome indexes for mapping with TopHat2. TopHat2 was run with default settings and the *–mate-inner-dist* and *–mate-std-dev* options set to 50 and 200 respectively. The mate-inner-distance parameter was estimated based on the fragment analyzer report.

**Table 2.** Overview of external programmes.

| Programme | Version |
| --- | --- |
| Blast | 2.2.29+ |
| BEDTools | 2.17.0 |
| Bowtie2 | 2.1.0 |
| Clustal Omega | 1.2.4 |
| Cufflinks | 2.1.1 |
| FastQC | 0.10.1 |
| Mcl | 14.137 |
| R | 3.0 and 3.1 |
| Ruby | 2.0 and 2.1 |
| SAMTools | 0.1.19 |
| TopHat2 | 2.0.11 |
| ViennaRNA | 2.1.8 |

**Table 3.** Ensembl genome versions and annotation files for each species.

| Species | Genome | Annotation |
|---|---|---|
| Opossum | monDom5 | ensembl release 75, feb 2014 |
| Mouse | mm10 | ensembl release 75, feb 2014 |
| Rat | rn5 | ensembl release 75, feb 2014 |
| Rhesus macaque | rheMac2 | ensembl release 77, oct 2014 |
| Human | hg38 | ensembl release 77, oct 2014 |

## Analysis of unmapped reads

We developed a custom pipeline to detect circRNAs (*Figure 1—figure supplement 1*), which performs the following steps: Unmapped reads with a phred quality value of at least 25 are used to generate 20 bp anchor pairs from the terminal 3' and 5'-ends of the read. Anchors are remapped with bowtie2 on the reference genome. Mapped anchor pairs are filtered for (1) being on the same chromosome, (2) being on the same strand and (3) for having a genomic mapping distance to each other of a maximum of 100 kb. Next, anchors are extended upstream and downstream of their mapping locus. They are kept if pairs are extendable to the full read length. During this procedure, a maximum of two mismatches is allowed. For paired-end sequencing reads, the mate read not mapping to the backsplice junction can often be mapped to the reference genome without any problem. However, it will be classified as 'unmapped read' (because its mate read mapping to the backsplice junction was not identified by the standard procedure). Next, all unpaired reads are thus selected from the accepted_hits.bam file generated by TopHat2 (singletons) and assessed for whether the mate read (second read of the paired-end sequencing read) of the anchor pair mapped between the backsplice coordinates. All anchor pairs for which (1) the mate did not map between the genomic backsplice coordinates, (2) the mate mapped to another backsplice junction or (3) the extension procedure could not reveal a clear breakpoint are removed. Based on the remaining candidates, a backsplice index is built with bowtie2 and all reads are remapped on this index to increase the read coverage by detecting reads that cover the BSJ with less than 20 bp, but at least 8 bp. Candidate reads that were used to build the backsplice index and now mapped to another backsplice junction are removed. Upon this procedure, the pipeline provides a first list of backsplice junctions. The set of scripts, which performs the identification of putative BSJs, as well as a short description of how to run the pipeline are deposited in the Git Repository ncSplice_circRNAdetection (https://github.com/Frenzchen/ncSplice_circRNAdetection; *Gruhl, 2017*).

## Trimming of overlapping reads

Due to small DNA repeats, some reads are extendable to more than the original read length. Therefore, overlapping reads were trimmed based on a set of canonical and non-canonical splice sites. For the donor site GT, GC, AT, CT were used and for the acceptor splice site AG and AC. The trimming is part of our custom pipeline described above, and the step will be performed automatically if the scripts are run.

## Generation of high confidence circRNA candidates from the comparison of RNase R-treated vs. -untreated samples

The detection of circRNAs relies on the identification of BSJs. These are, however, often only covered by a low number of reads, which carries considerable risk of mistaking biological or technical noise for a real circRNA event. Their circular structure makes circRNAs resistant to RNase R treatment – a feature that is not generally expected for spurious RNA molecules that are linear but may nevertheless resemble BSJs. We therefore compared BSJs between RNase R-treated and -untreated samples and determined whether BSJs detected in an untreated sample are enriched in the RNase R-treated sample. To generate a high-confidence dataset of circRNA candidates from the comparison of untreated and treated samples (*Figure 1—figure supplement 1*), we applied the following filtering steps (please also consult *Supplementary file 2* for a step-by-step description of filtering outcomes, using the mouse samples as an example.)

Filtering step 1 - mapping consistency of read pairs. When mapping paired-end sequencing data, both reads should ideally map to the genome (paired-end = 'pe'). However, in some cases, one of the mate reads cannot be mapped due to the complexity of the genomic locus. These reads are reported as 'singletons' ('se'). For each potential BSJ, we thus analysed the mapping behaviour of both read mates. BSJs for which read pairs in the untreated and RNase R-treated sample of the same biological replicate mapped both either in 'pe' or 'se' mode were kept; BSJs for which for example a read pair mapped in 'pe' mode in the untreated biological sample, but in 'se' mode in the RNase R-treated sample of the same biological replicate (and vise versa) were considered weak candidates and removed. This filtering step removed approximately 1% of the total, unique BSJs detected (*Supplementary file 2*).

Filtering step 2 - presence of a BSJ in untreated samples. We hypothesized that for circRNAs to be functionally important, they should generally be expressed at levels that are high enough to make them detectable in the normal samples, that is without RNase R treatment. We thus removed all BSJs which were only present in RNase R-treated samples, but undetectable in any of the untreated, biological replicates (cut-off for absence/presence = minimum one read mapping to BSJ). This filtering step removed approximately 75% of the initially detected BSJs (*Supplementary file 2*).

Filtering step 3 - enrichment after RNase R treatment. RNase R treatment leads to the enrichment of BSJs in the total number of detected junctions due to the preferential degradation of linear RNAs. To calculate the enrichment factor, BSJs were normalised by the size factor (as described in Materials and methods, section *Reconstruction of circRNA isoforms*) of each sample and the mean normalised count was calculated for each condition (untreated and RNase R-treated). Next, the log2-enrichment for RNase R-treated vs. -untreated samples was calculated. All BSJs for which the log2-enrichment was below 1.5 were removed. This filtering step removed another 15% of the originally detected unique BSJs (*Supplementary file 2*).

Filtering step 4 - minimum expression levels. CPM (counts per million) values for BSJs were calculated for each tissue as follows:

$$counts = \frac{counts\_rep1 + counts\_rep2 + counts\_rep3}{3}$$

$$totalMappedReads = \frac{mappedReads\_rep1 + mappedReads\_rep2 + mappedReads\_rep3}{3}$$

$$CPM = \frac{counts \cdot 10^6}{totalMappedReads}$$

All BSJs with at least 0.05 CPM were kept. These loci were considered strong circRNA candidates and used for all subsequent analyses. After this final filtering step, less than 1% of the original BSJs are left (*Supplementary file 2*).

## Manual filtering steps

We observed several genomic loci in rhesus macaque and human that were highly enriched in reads for putative BSJs (no such problem was detected for opossum, mouse and rat). Manual inspection in the UCSC genome browser indicated that these loci are highly repetitive. The detected BSJs from these regions probably do not reflect BSJs, but instead issues in the mapping procedure. These candidates were thus removed manually; the concerned regions are listed in *Table 4*.

All following analyses were conducted with the circRNA candidates that remained after this step.

## Reconstruction of circRNA isoforms

To reconstruct the exon structure of circRNA transcripts in each tissue, we made use of the junction enrichment in RNase R treated samples. To normalise junction reads across libraries, the size factors based on the geometric mean of common junctions in untreated and treated samples were calculated as

$$geometric\_mean = \left(\prod x\right)^{\frac{1}{length(x)}}$$

$$size\_factor = median\left(\frac{x}{geometric\_mean}\right)$$

with *x* being a vector containing the number of reads per junction. We then compared read

**Table 4.** Removed regions during mapping.

| Species | Tissue | Chromosome | Start | Stop | Strand |
|---|---|---|---|---|---|
| Rhesus macaque | Testis | 7 | 164261343 | 164283671 | + |
| Rhesus macaque | Testis | 7 | 22010814 | 22092409 | - |
| Rhesus macaque | Testis | 19 | 52240850 | 52288425 | - |
| Rhesus macaque | Testis | 19 | 59790996 | 59834798 | + |
| Rhesus macaque | Testis | 19 | 59790996 | 59847609 | + |
| Human | Testis | 2 | 178535731 | 178600667 | + |
| Human | Testis | 7 | 66429678 | 66490107 | - |
| Human | Testis | 9 | 97185441 | 97211487 | - |
| Human | Testis | 12 | 97492460 | 97561047 | + |
| Human | Testis | 14 | 100913431 | 100949596 | + |
| Human | Testis | 18 | 21765771 | 21849388 | + |

coverage for junctions outside and inside the BSJ for each gene and used the log2-change of junctions outside the backsplice junction to construct the expected background distribution of change in junction coverage upon RNase R treatment. The observed coverage change of junctions inside the backsplice was then compared to the expected change in the background distribution and junctions with a log2-change outside the 90% confidence interval were assigned as circRNA junctions; a loose cut-off was chosen, because involved junctions can show a decrease in coverage if their linear isoform was present at high levels before (degradation levels of linear isoforms do not correlate with the enrichment levels of circRNAs). Next, we reconstructed a splicing graph for each circRNA candidate, in which network nodes are exons connected by splice junctions (edges) (*Heber et al., 2002*). Connections between nodes are weighted by the coverage in the RNase R-treated samples. The resulting network graph is directed (because of the known circRNA start and stop coordinates), acyclic (because splicing always proceeds in one direction), weighted and relatively small. We used a simple breadth-first-search algorithm to traverse the graph and to define the strength for each possible isoform by its mean coverage. Only the strongest isoform was considered for all subsequent analyses.

## Reconstruction and expression quantification of linear mRNAs

We reconstructed linear isoforms based on the pipeline provided by *Trapnell et al., 2012* (Cufflinks + Cuffcompare + Cuffnorm). Expression levels were quantified based on fragments per million mapped reads (FPKM). Cufflinks was run per tissue and annotation files were merged across tissues with Cuffcompare. Expression was quantified with Cuffnorm based on the merged annotation file. All programs were run with default settings. FPKM values were normalised across species and tissues using a median scaling approach as described in *Brawand et al., 2011*.

## Identification of shared circRNA loci between species

### Definition and identification of shared circRNA loci

Shared circRNA loci were defined on three different levels depending on whether the 'parental gene', the 'circRNA locus' in the gene or the 'start/stop exons' overlapped between species (see *Figure 2A* and *Figure 2—figure supplement 1A*). Overall considerations of this kind have recently also been outlined in *Patop et al., 2019*.

Level 1 - Parental genes: One-to-one (1:1) therian orthologous genes were defined between opossum, mouse, rat, rhesus macaque and human using the Ensembl orthology annotation (confidence intervals 0 and 1, restricted to clear one-to-one orthologs). The same procedure was performed to retrieve the 1:1 orthologous genes for the eutherians (mouse, rat, rhesus macaque, human), for rodents (mouse, rat), and primates (rhesus macaque, human). Shared circRNA loci between species were assessed by counting the number of 1:1 orthologous parental genes between the five species. The analysis was restricted to protein-coding genes.

Level 2 - circRNA locus: To identify shared circRNA loci, all circRNA exon coordinates from a given gene were collapsed into a single transcript using the *bedtools merge* option from the BED-Tools toolset with default options. Next, we used liftOver to compare exons from the collapsed transcript between species. The minimal ratio of bases that need to overlap for each exon was set to 0.5 (*-minMatch=0.5*). Collapsed transcripts were defined as overlapping between different species if they shared at least one exon, independent of the exon length.

Level 3 - start/stop exon: To identify circRNAs sharing the same first and last exon between species, we lifted exons coordinates between species (same settings as described above, *liftOver, -minMatch=0.5*). The circRNA was then defined as 'shared', if both exons were annotated as start and stop exons in the respective circRNAs of the given species. Note, that this definition only requires an overlap for start and stop exons, internal circRNA exons may differ.

Given that only circRNAs that comprise corresponding (1:1 orthologous exons) in different species might at least potentially and reasonably considered to be homologous (i.e. might have originated from evolutionary precursors in common ancestors) and the Level 3 definition might require strong evolutionary conservation of splice sites (i.e. with this stringent definition many shared loci may be missed), we decided to use the level 2 definition (circRNA locus) for the analyses presented in the main text, while we still provide the results for the Level 1 and 3 definitions in the supplement (*Figure 2—figure supplement 1A*). Importantly, defining shared circRNA loci at this level allows us to also compare circRNA hostspots which have been defined using a similar classification strategy.

## Clustering of circRNA loci between species

Based on the species set in which shared circRNA loci were found, we categorised circRNAs in the following groups: species-specific, rodent, primate, eutherian, and therian circRNAs. To be part of the rodent or primate group, the circRNA has to be expressed in both species of the lineage. To be part of the eutherian group, the circRNA has to be expressed in three species out of the four species mouse, rat, rhesus macaque and human. To be part of the therian group, the circRNA needs to be expressed in opossum and in three out of the four other species. Species-specific circRNAs are either present in one species or do not match any of the other four categories. The usage of multiple species for defining shared loci, allowed to define 'mammalian circRNAs' with high confidence (*Figure 2—figure supplement 1B*). To define the different groups, we used the cluster algorithm MCL (*Enright et al., 2002*; *Dongen, 2000*). MCL is frequently used to reconstruct orthology clusters based on blast results. It requires input in *abc* format (file: *species.abc*), in which *a* corresponds to event a, *b* to event b and a numeric value *c* that provides information on the connection strength between event a and b (e.g. blast p-value). If no p-values are available as in this analysis, the connection strength can be set to 1. MCL was run with a cluster granularity of 2 (*option -I*).

```
$ mcxload -abc species.abc –stream-mirror -o species.mci -write-tab species.tab
$ mcl species.mci -I 2
$ mcxdump -icl out.species.mci.I20 -tabr species.tab -o dump.species.mci.I20
```

## PhastCons scores

Codings exons were selected based on the attribute 'transcript_biotype = protein_coding' in the gtf annotation file of the respective species and labelled as circRNA exons if they were in our circRNA annotation. Exons were further classified into UTR-exons and non-UTR exons using the ensembl field 'feature = exon' or 'feature = UTR'. Since conservation scores are generally lower for UTR-exons (*Pollard et al., 2010*), any exon labelled as UTR-exon was removed from further analyses to avoid bias when comparing circRNA and non-circRNA exons. Genomic coordinates of the remaining exons were collapsed using the *merge* command from the BEDtools toolset (*bedtools merge input_file -nms -scores collapse*) to obtain a list of unique genomic loci. PhastCons scores for all exon types were calculated using the conservation scores provided by the UCSC genome browser (mouse: phastCons scores based on alignment for 60 placental genomes; rat: phastCons scores based on alignment for 13 vertebrate genomes; human: phastCons scores based on alignment for 99 vertebrate genomes). For each gene type (parental or non-parental), the median phastCons score was calculated for each exon type within the gene (if non-parental: median of all exons; if parental: median of exons contained in the circRNA and median of exons outside of the circRNA).

## Tissue specificity of exon types

Using the DEXseq package (from HTSeq 0.6.1), reads mapping on coding exons of the parental genes were counted. The exon-bins defined by DEXseq (filtered for bins >=10 nt) were then mapped and translated onto the different exon types: UTR-exons of parental genes, exons of parental genes that are not in a circRNA, circRNA exons. For each exon type, an FPKM value based on the exon length and sequencing depth of the library was calculated.

$$FPKM = \frac{counts\_for\_exon\_type \cdot 10^9}{exon\_type\_length / sequencing\_depth}$$

Exons were labelled as expressed in a tissue, if the calculated FPKM was at least 1. The maximum number of tissues in which each exon occurred was plotted separately for UTR-exons, exons outside the circRNA and contained in it.

## GC amplitude

The ensembl annotation for each species was used to retrieve the different known transcripts in each coding gene. For each splice site, the GC amplitude was calculated using the last 250 intronic bp and the first 50 exonic bp (several values for the last $n$ intronic bp and the first $m$ exonic bp were tested beforehand, the 250:50 ratio was chosen, because it gave the strongest signal). Splice sites were distinguished by their relative position to the circRNA (flanking, inside or outside). A one-tailed and paired Mann-Whitney U test was used to assess the difference in GC amplitude between circRNA-related splice sites and others.

## Definition of highly expressed circRNAs

For each species and tissues, circRNAs were grouped into lowly expressed and highly expressed circRNAs based on whether they were found below or above the 90% expression quantile of the respective tissue. Candidates from different tissues were then merged to obtain a unique list of highly expressed circRNAs for each species.

## Parental gene analysis

### GC content of exons and intron

The ensembl annotation for each species was used to retrieve the different known transcripts in each coding gene. Transcripts were collapsed per-gene to define the exonic and intronic parts. Introns and exons were distinguished by their relative position to the circRNA (flanking, inside, or outside). The GC content was calculated based on the genomic DNA sequence. On a per-gene level, the median GC content for each exon and intron type was used for further analyses. Differences between the GC content were assessed with a one-tailed Mann-Whitney U test.

### Gene self-complementarity

The genomic sequence of each coding gene (first to last exon) was aligned against itself in sense and antisense orientation using megaBLAST with the following call:

```
$ blastn -query seq.fa -subject seq.fa -task dc-megablast -word_size 12 -outfmt
"6 qseqid qstart qend sseqid sstart send sstrand length pident nident mismatch
bitscore evalue" > blast.out
```

The resulting alignments were filtered for being purely intronic (no overlap with any exon). The fraction of self-complementarity was calculated as the summed length of all alignments in a gene divided by its length (first to last exon).

### Generalised linear models

All linear models were developed in the R environment. The presence of multicollinearity between predictors was assessed using the *vif()* function from the R package *car* (version 3.0.3) to calculate the variance inflation factor. Predictors were scaled to be able to compare them with each other using the *scale()* function as provided in the R environment.

For parental genes, the dataset was split into training (80%) and validation set (20%). To find the strongest predictors, we used the R package *bestglm* (version 0.37). Each model was fitted on the complete dataset using the command *bestglm()* with the information criteria set to 'CV' (CV = cross validation) and the number of repetitions *t = 1000*. The model family was set to 'binomial' as we were merely interested in predicting the presence (1) or absence (0) of a parental gene. Significant predictors were then used to report log-odds ratios and significance levels for the validation set using the default *glm()* function of the R environment. Log-odds ratios, standard errors and confidence intervals were standardised using the *beta()* function from the *reghelper* R package (version 1.0.0) and are reported together with their p-values in *Supplementary file 6*. The same approach was used to predict which parental genes are likely to be a circRNA hotspot with the only difference that the underlying data was filtered for parental genes. All parental genes were then analysed for the presence (1) or absence (0) of a hotspot. Log-odds ratios, standard errors, and confidence intervals are reported together with their p-values in *Supplementary file 8*.

For the correlation of hotspot presence across the number of species, a generalised linear model was applied using the categorical predictors 'lineage' (=circRNA loci shared within rodents or primates), 'eutherian' (=circRNA loci shared within rodents and primates) and 'therian' (=circRNA loci shared within opossum, rodents, and primates). Log-odds ratios, standard errors, and confidence intervals were standardised using the *beta()* function from the *reghelper* R package (version 1.0.0) and are reported together with their p-values in *Supplementary file 7*.

## Comparison to human and mouse circRNA heart dataset

The circRNA annotations for human and mouse heart as provided by *Werfel et al., 2016* were, based on the parental gene ID, merged with our circRNA annotations. Prediction values for parental genes were calculated using the same general linear regression models as described above (section *Generalised linear models* in Materials and methods) with genomic length, number of exons, GC content, expression levels, reverse complements (RVCs), and phastCons scores as predictors. Prediction values were received from the model and compared between parental genes predicted by our and the Werfel dataset as well as between the predictors in non-parental and parental genes of the Werfel dataset (*Figure 3—figure supplement 4*).

## Integration of external studies

1. Replication time
   Values for the replication time were used as provided in *Koren et al., 2012*. Coordinates of the different replication domains were intersected with the coordinates of coding genes using BEDtools (*bedtools merge -f 1*). The mean replication time of each gene was used for subsequent analyses.
2. Gene expression steady-state levels
   Gene expression steady-state levels and decay rates were used as provided in Table S1 of *Pai et al., 2012*.
3. GHIS
   Genome-wide haploinsufficiency scores for each gene were used as provided in Supplementary Table S2 of *Steinberg et al., 2015*.

## Repeat analyses

### Generation of length- and GC-matched background dataset

Flanking introns were grouped into a matrix of *i* columns and *j* rows representing different genomic lengths and GC content; *i* and *j* were calculated in the following way:

$$i = seq(from = quantile(GCcontent, 0.05), to = quantile(GCcontent, 0.95), by = 0.01)$$
$$j = seq(from = quantile(length, 0.05), to = quantile(length, 0.95), by = 1000)$$

Flanking introns were sorted into the matrix based on their GC content and length. A second matrix with the same properties was created containing all introns of coding genes. From the latter, a submatrix was sampled with the same length and GC distribution as the matrix for flanking introns. The length distribution and GC distribution of the sampled introns reflect the distributions for the flanking introns as assessed by a Fisher's t Test that was non-significant.

## Repeat definition

The RepeatMasker annotation for full and nested repeats were downloaded for all genomes using the UCSC Table browser (tracks 'RepeatMasker' and 'Interrupted Rpts') and the two files merged. Nested repeats were included, because it was shown that small repetitive regions are sufficient to trigger base pairing necessary for backsplicing (*Liang and Wilusz, 2014*; *Kramer et al., 2015*). For rhesus macaque, the repeat annotation was only available for the rheMac3 genome. RVC coordinates were thus lifted from rheMac2 to rheMac3 (*liftOver, -minMatch=0.5*), which led to a significant drop of overlapping repeats and RVCs in comparison to the other species (only ~20% of RVCs could be intersected with an annotated repeat). The complete list of full and nested repeats was then intersected (*bedtools merge -f1*) with the above defined list of background and flanking introns for further analyses.

## Identification of repeat dimers

The complementary regions (RVCs) that were defined with megaBLAST as described above, were intersected with the coordinates of individual repeats from the RepeatMasker annotation. To be counted, a repeat had to overlap with at least 50% of its length with the region of complementarity (*bedtools merge -f 0.5*). As RVCs can contain several repeats, the 'strongest' dimer was selected based on the number of overlapping base pairs (=longest overlapping dimer).

We observed that the same genomic repeat can often be present in multiple RVCs. Assuming that repeats are unlikely to form multiple active dimers in the genome at the same given time point, we decided to correct dimer frequency for this 'co-counting' to not inflate our numbers and bias subsequent analyses (see also *Figure 5—figure supplement 2*). We calculated an overestimation factor based on the number of possible interactions each repeat had. Dimer frequency was then calculated as;

$$overestimation\_factor = \frac{co-counts_{\text{Repeat1}} + co-counts_{\text{Repeat2}}}{2}$$
$$dimer\_count_{\text{correct}} = \frac{dimer\_count}{overestimation\_factor}$$

The 'dimer list' obtained from this analysis for each species was further ranked according to the absolute frequency of each dimer. The proportion of the top-5 dimer frequency to all detected dimers, was calculated based on this list ($n_{\text{top}-5}$ / $n_{\text{all\_dimers}}$).

## Pairing scores of repeat dimers

Pairing scores for each TE class (based on the TE reference sequence) were defined by taking into account the (1) phylogenetic distance to other repeat families in the same species and (2) its binding affinity (the Minimal Free Energy = MFE of the dimer structure) to those repeats. We decided to not include the absolute TE frequency into the pairing score, because it is a function of the TE's age, its amplification and degradation rates. Simulating the interplay between these three components is not in scope of this study, and the integration of the frequency into the pairing score creates more noise as tested via PCA analyses (variance explained drops by 10%).

(1) Phylogenetic distance: TE reference sequences were obtained from Repbase (*Bao et al., 2015*) and translated into fasta-format for alignment (*reference_sequences.fa*). Alignments were then generated with Clustal Omega (v1.2.4) (*Sievers et al., 2011*) using the following settings:

```
$ clustalo -i reference_sequences.fa –distmat-out = repeats.mat –guidetree-out = repeats.dnd –full
```

The resulting distance matrix for the alignment was used for the calculation of the pairing score. Visualisation of the distance matrix (*Figure 4C*, *Figure 4—figure supplement 2*) was performed using the standard R functions *dist(method="euclidian')* and *hclust(method="ward.D2')*. Since several TE classes evolved independently from each other, the plot was manually modified to remove connections or to add additional information on the TE's origin from literature.

(2) Binding affinity: To estimate the binding affinity of individual TE dimers, the free energy of the secondary structure of the respective TE dimers was calculated with the RNAcofold function from the ViennaRNA Package:

```
$ RNAcofold −a −d2 < dimerSequence.fa
```

with *dimerSequence.fa* containing the two TE reference sequences from which the dimer is composed. The resulting MFE values were used to calculate the pairing score.

(3) Final pairing score: To generate the final pairing score, values from the distance matrix and the binding affinity were standardised (separately from each other) to values between 0 and 1:

$$f(x) = \frac{x - min(v)}{max(v) - min(v)}$$

with *x* being the pairing affinity/dimer frequency and *minv* and *maxv* the minimal and maximal observed value in the distribution. The standardised values for the binding affinity and dimer frequency were then summed up (=pairing score) and classified by PCA using the R environment:

```
$ pca <- prcomp(score, center=TRUE, scale.=FALSE)
```

PC1 and PC2 were used for subsequent plotting with the absolute frequency of dimers represented by the size of the data points (*Figure 4D–F*, *Figure 4—figure supplement 2*).

## Dimer composition in shared and species-specific circRNA loci

Dimers were sorted by their frequency in all parental genes and the 100 most and least frequent dimers were selected to be analysed for their enrichment in shared vs. species-specific circRNA loci. The two dimer frequency distributions were compared using a Wilcoxon Signed Rank Test. Dimer age was defined on whether the repeat family originated in a given species (=rank 1), lineage (=rank 2), in all eutherian species of this study (=rank 3) or all therian species (=rank 4). Since a dimer is composed of two repeats, the 'mean dimer age' based on the rank value was taken. Based on this analysis, the top-5 most frequent and enriched dimers were then defined.

## Calculation of TE degradation levels

We analysed repeat degradation levels for all TEs present in the top-5 dimers of each species. RepeatMasker annotations were downloaded from the UCSC Table browser for all genomes (see Materials and methods, section *Repeat definition*). The milliDiv values for each TE were retrieved from this annotation for full and nested repeats. All indivudal TEs were then grouped as 'species-specific' or 'shared' based on whether the circRNA parental gene produced species-specific or shared circRNA loci. Significance levels for milliDiv differences between the TE groups were assessed with a simple Mann-Whitney U test.

## Binding affinity of dimers

The binding affinity of dimers was calculated with the RNAcofold function from the ViennaRNA Package:

```
$ RNAcofold −a −d2 < dimerSequence.fa
```

with *dimerSequence.fa* containing the two TE genomic sequences from which the dimer is composed. To reduce calculation time for human and opossum, the analysis was restricted to the respective top-5 dimers (see section *Dimer composition in shared vs. species-specific circRNA loci*). For each gene of the two groups (shared/species-specific), the least degraded dimer based on its mean milliDiv value was chosen. Filtering based on the least degraded dimer, let to a strong enrichment of only a subset of the top-5 dimers in each species. If enough observations for a statistical test were present, the two distributions (shared/species-specific) were compared using a Student's t-Test.

## Acknowledgements

We thank the Lausanne Genomics Technologies Facility for high-throughput sequencing support; Jean Halbert, Delphine Valloton and Angelica Liechti for opossum, mouse and rat tissue dissection and RNA extractions; Philipp Khaitovich for providing human and rhesus macaque samples; Bulak Arpat and Thomas O Auer for discussions on the manuscript; and Ioannis Xenarios for discussion and support with IT-infrastructure and data archiving. This research was supported by grants from the Swiss National Science Foundation to DG (NCCR RNA & Disease and individual grant 179190) and from the European Research Council to HK (242597, SexGenTransEvolution; and 615253, Onto-TransEvol). FG was supported by the SIB PhD Fellowship granted by the Swiss Institute of Bioinformatics and the Fondation Leenaards. PJ was supported by Human Frontiers Science Program long-term fellowship LT000158/2013 L.

## Additional information

### Funding

| Funder | Grant reference number | Author |
|---|---|---|
| Swiss Institute of Bioinformatics | SIB PhD Fellowship | Franziska Gruhl |
| Human Frontier Science Program | LT000158/2013-L | Peggy Janich |
| European Research Council | 242597 (SexGenTransEvolution) | Henrik Kaessmann |
| European Research Council | 615253 (OntoTransEvol) | Henrik Kaessmann |
| Swiss National Science Foundation | NCCR RNA & Disease (141735) | David Gatfield |
| Swiss National Science Foundation | NCCR RNA & Disease (182880) | David Gatfield |
| Swiss National Science Foundation | individual grant 179190 | David Gatfield |

The funders had no role in study design, data collection and interpretation, or the decision to submit the work for publication.

### Author contributions

Franziska Gruhl, Conceptualization, Software, Formal analysis, Validation, Investigation, Visualization, Methodology, Writing - original draft, Writing - review and editing; Peggy Janich, Conceptualization, Investigation, Methodology; Henrik Kaessmann, Conceptualization, Supervision, Funding acquisition, Writing - review and editing; David Gatfield, Conceptualization, Supervision, Funding acquisition, Methodology, Writing - original draft, Writing - review and editing

### Author ORCIDs

Franziska Gruhl https://orcid.org/0000-0002-2613-7211
Peggy Janich https://orcid.org/0000-0003-1045-7365
Henrik Kaessmann https://orcid.org/0000-0001-7563-839X
David Gatfield https://orcid.org/0000-0001-5114-2824

### Ethics

Human subjects: The human post-mortem samples were provided by the NICHD Brain and Tissue Bank for Developmental Disorders at the University of Maryland (USA). They originated from individuals with diverse causes of death that, given the information available, were not associated with the organ sampled. Written consent for the use of human tissues for research was obtained from all donors or their next of kin by this tissue bank. The use of these samples was approved by an ERC Ethics Screening panel (associated with HK's ERC Consolidator Grant 615253, OntoTransEvol), and, in addition, by the local ethics committee in Lausanne (authorization 504/12).

Animal experimentation: Mouse samples were collected by the Kaessmann lab at the Center for Integrative Genomics in Lausanne. Rat samples were kindly provided by Carmen Sandi, EPFL, Lausanne. Opossum samples were kindly provided by Peter Giere, Museum für Naturkunde, Berlin. All animal procedures were performed in compliance with national and international ethical guidelines and regulations for the care and use of laboratory animals and were approved by the local animal welfare authorities (Vaud Cantonal Veterinary office, Berlin State Office of Health and Social Affairs). The rhesus macaque samples were provided by the Suzhou Experimental Animal Center (China); the Biomedical Research Ethics Committee of Shanghai Institutes for Biological Sciences reviewed the use and care of the animals in the research project (approval ID: ER-SIBS-260802P). All rhesus macaques used in this study suffered sudden deaths for reasons other than their participation in this study and without any relation to the organ sampled. The use of all samples for the work described in this study was approved by an ERC Ethics Screening panel (associated with HK's ERC Consolidator Grant 615253, OntoTransEvol).

### Decision letter and Author response

Decision letter https://doi.org/10.7554/eLife.67991.sa1
Author response https://doi.org/10.7554/eLife.67991.sa2

## Additional files

### Supplementary files

• Supplementary file 1. Sample overview. Summary of organism, tissue, age and sex for each sample; last column shows the RNA Quality Number (RQN) for the extracted RNA.

• Supplementary file 2. Filtering steps and reduction of circRNAs candidates during the identification pipeline. Description of the different filtering steps applied to generate a high confidence circRNA dataset based on the comparison of untreated and RNase R-treated samples. The number of unique BSJs left after each filtering step is shown for each tissue (see Materials and methods, section *Generation of high confidence circRNA candidates from the comparison of RNase R-treated vs. -untreated samples*); mouse was chosen as representative example.

• Supplementary file 3. Detected back splice junctions (BSJs) across samples. Table summarises the total number of detected BSJs after the filtering step in each species. The percentage of BSJs that are unique to one, two, three or more than three samples of the same species is shown.

• Supplementary file 4. Total number of circRNAs in different species and tissues. Indicated is the total number of different circRNAs that were annotated in each of the tissues across species.

• Supplementary file 5. Mean amplitude correlations. Spearman's rank correlation for the GC amplitude and GC content of introns and exons are calculated for each isochore and species. The mean correlation between the GC amplitude and GC content of introns and exons is shown for different splice sites relative to the circRNA.

• Supplementary file 6. GLM summary for presence of parental genes. A generalised linear model was fitted to predict the probability of coding genes to be a parental gene ($n_{opossum}$ = 18807, $n_{mouse}$ = 22015, $n_{rat}$ = 11654, $n_{rhesus}$ = 21891, $n_{human}$ = 21744). The model was trained on 80% of the data (scaled values, cross-validation, 1000 repetitions, shown in rows labeled as 'prediction'). Only the best predictors were kept and then used to predict probabilities for the remaining 20% of data points (validation set, shown in rows labeled as 'validation'). Log-odds ratios, standard error and 95% confidence intervals (CI) for the validation set have been (beta) standardised.

• Supplementary file 7. GLM summary for 'sharedness' of hotspots. A generalised linear model was fitted to predict the probability of a hotspot to be present across multiple species ($n_{opossum}$ = 872, $n_{mouse}$ = 848, $n_{rat}$ = 665, $n_{rhesus}$ = 1682, $n_{human}$ = 2022). Reported log-odds ratios, standard error and 95% confidence intervals (CI) are (beta) standardised.

• Supplementary file 8. GLM summary for circRNA hotspots among parental genes. A generalised linear model was fitted to predict the probability of circRNA hotspots among parental genes; parental genes were filtered for circRNAs that were either species-specific or occurred in orthologous loci

across therian species ($n_{opossum}$ = 869, $n_{mouse}$ = 503, $n_{rat}$ = 425, $n_{rhesus}$ = 912, $n_{human}$ = 1213). The model was trained on 80% of the data (scaled values, cross-validation, 1000 repetitions, shown in rows labeled as 'prediction'). Only the best predictors were kept and then used to predict probabilities for the remaining 20% of data points (validation set, shown in rows labeled as 'validation'). Log-odds ratios, standard error and 95% confidence intervals (CI) for the validation set have been (beta) standardised.

• Supplementary file 9. Analysis of highly expressed circRNAs. Highly expressed circRNAs were defined as the circRNAs present in the 90% expression quantile of a tissue in a species. Per species, the circRNAs in the 90% expression quantiles from each of the three tissues were then pooled for further analysis ($n_{opossum}$ = 158, $n_{mouse}$ = 156, $n_{rat}$ = 217, $n_{rhesus}$ = 340, $n_{human}$ = 471) and their properties compared to circRNAs outside the 90% expression quantile. Highly expressed circRNAs are designated '1', others '0'. Differences in genomic length, circRNA length, exon number and GLM model performance were assessed with a Student's t-Test; p-values are indicated in the table (ns = non-significant).

• Supplementary file 10. GLM for highly expressed circRNAs based on 'age groups'. A generalised linear model was fitted on the complete dataset to predict the probability of parental genes of highly expressed circRNAs to be produce circRNAs in multiple species ($n_{opossum}$ = 869, $n_{mouse}$ = 844, $n_{rat}$ = 661, $n_{rhesus}$ = 1673, $n_{human}$ = 2016). The 'sharedness' definition is based on the phylogeny of species as: present in only one species, in rodents (mouse, rat) or primates (rhesus, human), eutherian species (rodents + at least one primate, or primates + at least one rodent) and therian species (opossum + rodents + at least one primate, or opossum + primates + at least one rodents). Log-odds ratios, standard error, 95% confidence intervals (CI) and p-values are shown.

• Supplementary file 11. Frequency and enrichment of top-5 dimers in shared and species-specific circRNA loci. The total number of detected top-5 dimers in shared and species-specific circRNA loci as well as their enrichment after correction for co-occurrence in multiple RVCs (see Materials and methods) are shown. Loci were normalised by the number of detected genes in each category before calculating the enrichment of dimers in shared over species-specific loci. The number of parental genes in both categories is shown below the species name. For mouse, only the top-3 dimers, which are outside the 95% frequency quantile, are shown (see Materials and methods). For rhesus, the analysis could only be done on a subset of genes due to lifting uncertainties between the rheMac2 and the rheMac3 genome (see Materials and methods).

• Supplementary file 12. CircRNA annotation file for opossum. A gtf-file with all circRNA transcripts including the transcript and exon coordinates.

• Supplementary file 13. CircRNA annotation file for mouse. A gtf-file with all circRNA transcripts including the transcript and exon coordinates.

• Supplementary file 14. CircRNA annotation file for rat. A gtf-file with all circRNA transcripts including the transcript and exon coordinates.

• Supplementary file 15. CircRNA annotation file for rhesus macaque. A gtf-file with all circRNA transcripts including the transcript and exon coordinates.

• Supplementary file 16. CircRNA annotation file for human. A gtf-file with all circRNA transcripts including the transcript and exon coordinates.

• Transparent reporting form

## Data availability

Sequencing data have been deposited in GEO under accession code GSE162152.

The following dataset was generated:

| Author(s) | Year | Dataset title | Dataset URL | Database and Identifier |
|---|---|---|---|---|
| Gruhl F, Janich P, Kaessmann H, Gatfield D | 2021 | Identification and evolutionary comparison of circular RNAs in five mammalian species and | https://www.ncbi.nlm.nih.gov/geo/query/acc.cgi?acc=GSE162152 | NCBI Gene Expression Omnibus, GSE162152 |

three organs.

The following previously published datasets were used:

| Author(s) | Year | Dataset title | Dataset URL | Database and Identifier |
|---|---|---|---|---|
| Koren A, Polak P, Nemesh J, Michaelson JJ, Sebat J, Sunyaev SR, McCarroll SA | 2012 | DNA replication time of the human genome G1 phase | https://www.ncbi.nlm.nih.gov/sra/SRX147697 [accn] | NCBI Sequence Read Archive, SRA052697 |
| Pai AA, Cain CE, Mizrahi-Man O, Leon S, Lewellen N, Veyrieras J-B, Degner JF, Gaffney DJ, Pickrell JK, Stephens M, Pritchard JK, Gilad Y | 2012 | The contribution of RNA decay quantitative trait loci to inter-individual variation in steady-state gene expression levels | https://www.ncbi.nlm.nih.gov/geo/query/acc.cgi?acc=GSE37451 | NCBI Gene Expression Omnibus, GSE37451 |
| Brawand D, Soumillon M, Necsulea A, Julien P, Csardi G, Harrigan P, Weier M, Liechti A, Aximu-Petri A, Kircher M, Albert FW, Zeller U, Khaitovich P, Grützner F, Bergmann S, Nielsen R, Pääbo S, Kaessmann H | 2011 | The evolution of gene expression levels in mammalian organs | https://www.ncbi.nlm.nih.gov/geo/query/acc.cgi?acc=GSE30352 | NCBI Gene Expression Omnibus, GSE30352 |

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
