## [Decision Letter]

**Acceptance summary:**

This manuscript proposes that conserved occurrence of circRNAs across species is the indirect consequence of convergent insertion of different, species-specific transposable elements and therefore possibly the by-product of independent, recent transposon insertions, rather than the product of natural selection acting on conserved functions of circRNAs. Thus, the paper offers a novel perspective for considering circRNAgenesis and evolution and is relevant to the question of whether most circRNAs have active functions or are by-products of transcriptional noise.

**Decision letter after peer review:**

Thank you for submitting your article "Circular RNA repertoires are associated with evolutionarily young transposable elements" for consideration by *eLife*. Your article has been reviewed by 3 peer reviewers, including Juan Valcárcel as Reviewing Editor and Reviewer #1, and the evaluation has been overseen by Detlef Weigel as the Senior Editor.

Summary

Gruhl et al., sequenced thousands of circular (RNase R-resistant) RNAs from 3 organs in 5 different species and carried out detailed analyses of the genomic loci involved in the generation of these transcripts, whose heterogeneity increases with sequence coverage, suggesting substantial noise, but limited conservation across species. They identify loci with high circRNA production (hotspots), which are enriched in conserved circRNAs, and features associated with them, including GC amplitudes and content, genomic length and exon number, expression in multiple tissues, conservation scores and sequence repetitiveness in the introns flanking exons undergoing circularization by back-splicing. The authors link the latter to enrichment in repetitive transposable elements (TEs) and the detailed characterization of these TEs in different species leads the authors to propose that conservation in circRNA production across species is the result of convergent evolution through independent insertion of evolutionarily young, species-specific TEs in the introns flanking the circularized exons.

This study tackles an interesting and timely topic, namely the extent to which circRNAs perform biological functions or are rather byproducts of transcriptional noise. While previous publications have linked circRNA biogenesis to transposable elements located in flanking introns, even in evolutionary terms (e.g. Ivanov et al., Cell Rep 2015, PMID: 25558066; Dong et al., RNA Biol 2017, PMID: 27982734; and Ji et al., Cell Rep 2019, PMID: 30893614), the manuscript provides novel perspectives on the evolutionary origin of circRNAs and can have general implications for understanding their biosynthesis and function.

While the authors build a convincing argument using the data they have generated, the number of circRNAs analyzed remain a small fraction of currently annotated circRNA datasets, which raises the question of how general is the model put forward by the authors. A second, complementary question is whether the model applies to highly abundant circRNAs -which may be the subset more enriched in functional circRNAs.

Essential revisions:

1. The authors collected three organs from five species for RNA sequencing to annotate circRNAs for their subsequent analysis, and obtained about 4,500 circRNAs in human and much fewer in other four species. However, significantly more circRNAs have been previously reported. for example, in some publicly-available circRNA databases, such as circAtlas (Wu et al., Genome Biol 2020, PMID: 32345360), circBase (Glažar et al., RNA 2014, PMID: 25234927), CIRCpedia (Zhang et al., Genome Res 2016, PMID: 27365365), more than 400,000 circRNAs are annotated from human samples. Very recently, two publications by using third generation sequencing technologies further confirmed a large number of circRNAs in human (including Xin et al., Nat Commun 2021, PMID: 33436621 and Zhang et al., Nat Biotechnol 2021, PMID: 33707777). It is not clear why the authors only used their own circRNA annotation for the current study, as this might lead to a significant bias in the downstream analysis (e.g. affecting the parental genes of circRNAs to be compared to genes that do not generate circRNAs).

2. A complementary question is whether the model put forward by the authors applies to highly abundant circRNAs, which are likely to be enriched in functional circRNAs, as the sense in the field is that only highly abundant circRNAs -particularly in the nervous system- might be functional. Can the authors partition their assessment of conservation for the different quantiles of expression? It is indeed possible that only circRNAs in the top quantile of expression will provide insights into the potential functional importance of circRNAs. The authors could also take the opposite approach: for those few circRNAs for which some functionality has been proven, are the pattern of evolution and elements flanking the circularizable exons similar to the ones they describe generally?

3. The authors identified circRNAs using RNAseR-treated samples. While RNAseR enriches generally for circRNAs, the authors don't have data reporting the abundance of the circRNAs in mock treated samples. Without a mock control is impossible for the authors to determine the actual resistance to the degradation by the exonuclease and hence determine which potential backsplicing junctions are really coming from circular molecules. The authors could do at least for some of the tissues/species show that the circRNAs they identified have been previously shown to be real (e.g., by a canonical comparison mock vs RNAseR treated samples).

4. For the analysis of enriched TEs of flanking introns of circRNAs or circRNAs shared across species, except the ages of TEs, other factors should be also included in these analyses. For example, whether pairing abilities of RVCs alone can explain the phenomenon? Whether the number of RVCs in the flanking introns is related to the conserved or species specific circRNAs? In addition, In this case, the general conclusion about circRNA biogenesis and TE insertion events can be biased.

Suggested text revisions:

a. The discussion should further elaborate the concept that at least for some circRNAs, their main functionality is to regulate/limit the expression of the host gene, hence the exact exons that form circRNAs might not be important. Therefore it would not so important which exon in particular makes the cirRNA or even which elements in the flanking introns are driving this process but rather the existence of this during evolution. This is relevant especially for those circRNAs for which the circular molecules are abundant in comparison with the linear mRNAs generated from the locus and there are at least some cases in the literature in which has been shown that production of the circRNA comes at the expense of the linear transcript.

b. Discuss the evolutionary implications of the observation that circRNAs in fruitfly is not driven by intronic pairing of repetitive/transposable elements (Westholm, et al., Cell Rep 2014, PMID: 25544350).

c. On page 6, the authors stated "Coding genes in rhesus macaque and human are characterised by a bimodal GC content distribution". The H3 category contains all genes with GC content > 52% while L2 category contains genes with 37-42% GC contents (Figure 3A). It is not correct to define it as bimodal GC content distribution when the group standard has a large difference.

d. More data are needed to sustain the statement "circRNA parental genes are characterised by low GC content and high sequence repetitiveness". Low GC content and high sequence repetitiveness may be the consequences of the long intron length of parental genes and/or the enrichment of repeat elements in long introns. More stringent conditions, like comparable intron lengths between circRNA and non-circRNA parent genes would need to be set up for this type of comparison.*Reviewer #1 (Recommendations for the authors):*

I would recommend the authors to consider the following revisions:

1. There seems to be some contradiction between the argument that circRNAs tend to be generated from constitutive, widely expressed exons and the statement on page 3 that circRNAs "showed considerable tissue-specificity".

2. The authors may want to elaborate further on the possible links between the preferred genomic architectures of circRNAs hotspots and the concept of exon definition (e.g. enrichment in long flanking introns and differences in GC content between exons and introns, as reported by Ast and colleagues).

3. The predictive model generated by the authors produces very impressive results. In addition to sequence complementarity, a number of protein factors have been shown to support the formation of stem-loop structures associated with back splicing, including hnRNP L, MBNL1 or PTB. The authors may be in an excellent position to assess whether the presence of potential binding sites for these proteins contributes evolutionarily to the generation of circRNAs (or subtypes of them). More specifically, does the inclusion of binding sites for these factors improve the performance of their predictive model?

4. Figure 4A: it would be good to provide some measurement of the statistical relevance of the enrichment scores observed.

5. Figure 5 and related discussion about evolutionary age of TEs associated with circRNA biogenesis: is it possible that dimers co-evolving over long periods of time accumulate mutations but retain -through compensatory changes- their capacity to form base pairing interactions leading to looping out and back-splicing?*Reviewer #2 (Recommendations for the authors):*

In this manuscript entitled "Circular RNA repertoires are associated with evolutionarily young transposable elements", Franziska Gruhl et al., applied a series of computational analyses to investigate the evolution of circRNAs. They first generated several RNA-seq samples from three organs in five species. Based on their analysis with these newly generated, but limited, samples, they concluded a negative correlation between "circRNA hotspot" and CPM threshold with some genomic features (including low GC content, long genomics length and so on) related to circRNA parental genes. By performing an RVC (reverse complements) analysis, authors showed enrichment of species-specific TEs in circRNA flanking introns and more evolutionarily young TEs enriched in flaking introns of circRNA loci shared across species. However, since most of their findings are obtained from limited datasets of circRNAs, the overall analyses in this current version are not strong enough to support their conclusion. In addition, some other publications have already suggested the correlation between circRNA biogenesis and TE, and compared circRNAs across (more) species to depict its evolutionary expression pattern (for example Ivanov et al., Cell Rep 2015, PMID: 25558066; Dong et al., RNA Biol 2017, PMID: 27982734; and Ji et al., Cell Rep 2019, PMID: 30893614), and thus the significancy and novelty of this current manuscript was unclearly described.

1. The main pitfall of the current manuscript is to use a relatively small size of samples for analysis, which could lead to misleading conclusions. The authors collected three organs from five species for RNA sequencing to annotate circRNAs for their subsequent analysis, and obtained about 4,500 circRNAs in human and much fewer in other four species. Strikingly, significantly more circRNAs have been previously reported (many references in the field). Specifically, in some publicly-available circRNA databases, such as circAtlas (Wu et al., Genome Biol 2020, PMID: 32345360), circBase (Glažar et al., RNA 2014, PMID: 25234927), CIRCpedia (Zhang et al., Genome Res 2016, PMID: 27365365), more than 400,000 circRNAs are annotated in human samples. Very recently, two publications by using third generation sequencing technologies further confirmed a large number of circRNAs in human (including Xin et al., Nat Commun 2021, PMID: 33436621 and Zhang et al., Nat Biotechnol 2021, PMID: 33707777). It is not clear why the authors only used their own circRNA annotation for the current study, and this limited number of circRNAs may lead to significant bias in the downstream analysis. For example, the number of circRNA parental genes or non-circRNA parental genes would be totally different with distinct sets of circRNAs.

2. For the analysis of enriched TEs of flanking introns of circRNAs or circRNAs shared across species, except the ages of TEs, other factors should be also included in these analyses. For example, whether pairing abilities of RVCs alone can explain the phenomenon? Whether the number of RVCs in the flanking introns is related to the conserved or species specific circRNAs? In addition, in an evolutionary view of circRNAs, it has been reported that circRNAs in fruitfly is not driven by intronic pairing of repetitive/transposable elements (Westholm, et al., Cell Rep 2014, PMID: 25544350). In this case, the general conclusion about circRNA biogenesis and TE insertion events can be biased.

3. At page 6, the authors stated "Coding genes in rhesus macaque and human are characterised by a bimodal GC content distribution". The H3 category contains all genes with GC content > 52% while L2 category contains genes with 37-42% GC contents (Figure 3A). It is not suitable to define it as bimodal GC content distribution when the group standard has a large difference.

4. More data are needed for the statement "circRNA parental genes are characterised by low GC content and high sequence repetitiveness". Low GC content and high sequence repetitiveness may be the consequences of the long intron length of parental genes and/or the enrichment of repeat elements in long introns. More stringent conditions, like comparable intron lengths between circRNA and non-circRNA parent genes, should be set up for this type of comparison.

The regulation of circRNA biogenesis can be complex. It should be careful to evaluate different conditions to draw a convincing conclusion. The major pitfall of this study is to use relatively small population of circRNAs for the evolutionary study, which may lead to biased findings. In addition, more stringent controls are required for analyses throughout the study. The Discussion part seems to be unusually long, containing lots of assumptions that are needed to be proven. Please shorten.*Reviewer #3 (Recommendations for the authors):*

In this manuscript, the authors address an interesting and timely topic: what is the evolutionary conservation pattern of circular RNAs (circRNAs). For doing so, the authors first utilize RNAseR-RNAseq to identify circRNAs in five mammalian species. They then determine their conservation utilizing different criteria and find several conserved and even predictive features of genes harboring circRNAs in these species. Interestingly, the authors discover that few circRNAs arise from orthologous exonic loci across the studied species. Moreover, the small conservation seems to arise from convergent evolution driven by young specie-specific transposable elements.

One of the strengths of the manuscript is the topic, general idea and some parts of the analysis, like the sections in which the authors identify features of genes hosting circRNAs (including the hot spot analysis). While the methodology while determining the evolution of circRNAs is also appropriate, it seems to me that the authors have oversimplified some of the issues and should consider additional possibilities and limitations of their own data. For example, the authors identified circRNAs using RNAseR-treated samples. While RNAseR enriches generally for circRNAs, the authors don't have any data of the abundance of the circRNAs in mock treated samples. This is problematic for two reasons: 1. Some of the potential circRNAs might not be bona fide circRNAs; 2. Probably most of the circRNAs identified and further analyzed are of very low abundance and more likely to be noise/irrelevant. In addition, the authors utilize all the potential circRNAs for their genomic features and evolutionary assessments. This is also problematic, as the general sense in the field is that only highly abundant circRNAs might be functional. In this sense, the suggestion that most circRNAs are transcriptional noise is not new. Last but not least, the manuscript does not take into account the possibility that at least for some circRNAs, their main functionality is to regulate/limit the expression of the host gene, hence the exact exons that form circRNAs might not be important.

The manuscript is interesting and timely. However, the conclusions are too broad and the specific issues need to be addressed experimentally and/or by doing additional analysis.

More specifically:

– The approach for circRNA detection is at least problematic. Without a mock control is impossible for the authors to determine the actual resistance to the degradation by the exonuclease and hence determine which potential backsplicing junctions are really coming from circular molecules. One thing the authors could do is at least for some of the tissues/species show that the circRNAs they identified have been previously showed to be real (e.g., by a canonical comparison mock vs RNAseR treated samples).

– Authors seems to ignore that in vivo studies focused on highly expressed circRNAs. Can the authors partition their assessment of conservation for the different quantiles of expression? Indeed, I would believe that only circRNAs in the top quantile of expression are the ones that will really give insights into the potential functional importance of circRNAs.

– Notion of noise for lowly expressed circRNAs is prevalent in the field and the key question is whether the circRNAs expressed in the neural system at high levels have function. So only on those circRNAs is relevant to determine evolutionary conservation. Indeed, the authors could take the opposite approach; for those few circRNAs for which some functionality been proven, are the pattern of evolution and elements flanking the circularizable exons similar to the ones they describe generally?

– The manuscript ignores potential cis regulatory mechanisms of circRNA biogenesis. One could argue that an important feature of a given locus is to produce circRNAs as a way to limit the amount of linear RNA. In this sense, it would not so important which exon in particular makes the cirRNA or even which elements in the flanking introns are driving this process but rather the existence of this during elevation. This is relevant especially for those circRNAs for which the circular molecules are abundant in comparison with the linear mRNAs generated from the locus and there are at least some cases in the literature in which has been shown that production of the circRNA comes at the expense of the linear transcript. The authors could include this also as a criterion for evolution and not rule it out based on some theoretical assumptions.

---

## [Author Response]

Essential revisions:1. The authors collected three organs from five species for RNA sequencing to annotate circRNAs for their subsequent analysis, and obtained about 4,500 circRNAs in human and much fewer in other four species. However, significantly more circRNAs have been previously reported. for example, in some publicly-available circRNA databases, such as circAtlas (Wu et al., Genome Biol 2020, PMID: 32345360), circBase (Glažar et al., RNA 2014, PMID: 25234927), CIRCpedia (Zhang et al., Genome Res 2016, PMID: 27365365), more than 400,000 circRNAs are annotated from human samples. Very recently, two publications by using third generation sequencing technologies further confirmed a large number of circRNAs in human (including Xin et al., Nat Commun 2021, PMID: 33436621 and Zhang et al., Nat Biotechnol 2021, PMID: 33707777). It is not clear why the authors only used their own circRNA annotation for the current study, as this might lead to a significant bias in the downstream analysis (e.g. affecting the parental genes of circRNAs to be compared to genes that do not generate circRNAs).

We thank the Reviewers for raising the issue – this is indeed an important point to clarify. It is true that the publications and circRNA databases cited by the Reviewers have reported on very high numbers of molecularly distinct circRNA molecules, whereas we base our study on circRNA counts that are in the low thousands.

1) The most important answer to why we “only used [our] own circRNA annotation for the current study” lies in the unavailability of comparable data from previous studies. It is correct that we could have retrieved hundreds of thousands of circRNA sequences that have been reported from human samples and cell lines in the literature, yet we would not have been able to obtain matched data from, for example, opossum or macaque organs. In our experience, it is key for clean, unbiased evolutionary analyses to generate datasets in parallel and with the same methodology. We therefore do not think that it would have been a good option to use the large published human datasets and compare them with data from the other species that we would anyway have to generate ourselves. Thus, the most straightforward solution consisted in acquiring all biological samples ourselves, subsequently allowing us to ensure the best possible consistency of organs and of the data production and analysis pipelines. In particular, (i) all organs originated from young, sexually mature male individuals; (ii) our highquality biological samples, RNA and library preparations are fully traceable; (iii) the data was produced for all samples in parallel using the most appropriate experimental protocols.

We also note, however, that we actually did analyse complete published datasets, too, in order to independently validate some of our findings – please see the analysis of the human-mouse-rat datasets from Werfel et al., 2016; now in Figure 3 – Figure supplement 4.

2) Second, why do we “only” report on a few thousand circRNAs, as compared to the published hundreds of thousands? To understand this difference, we need to elaborate slightly more. The first important point to make is that the publications cited by the reviewers – for example the two most recent ones by Xin et al., and by Zhang et al., both from 2021 – actually used relatively permissive conditions for circRNA detection and annotation. We will explain this just below using the publication by Xin et al., in addition, at the end of this reply to Essential Revision 1 (see “Supplementary Rebuttal Information”), we provide a table that analyzes data from the three databases mentioned by the reviewers (circAtlas, circBase, CIRCpedia) and that summarizes why we think that an inclusion of the reported circRNAs would not have been helpful for our study.

With regard to the example of Xin et al., one should first note that this particular study is based on RNase R-treated samples, without the use of additional mock-treated libraries. We, however, used both library types and the enrichment factor between treated vs. mock treated as one of the criteria to define high-confidence circRNAs (see Essential revision 3). Moreover, the study by Xin et al., is based exclusively on a cancer-derived cell line, HEK293. HEK293 cells are hypotriploid (64 chromosomes) and are cytogenetically not very stable (as compared to organ samples that are the basis of our study). Chromosomal instability can easily be envisioned to lead to exon rearrangements that would lead to de novo creation of BSJs and abnormal splicing events that are not biologically relevant in vivo. Next, Xin et al., prepared 6 libraries - i.e. 3 technical library replicates sequenced from 2 biological HEK293 cell samples - and reported on around 40’000-60’000 BSJs/circRNA isoforms per library. Of note, before applying our enrichment/filtering steps to remove noise/sporadic BSJ detection in our samples, the numbers of BSJs that we detect are within the same order of magnitude (see Supplementary File 2). Please also note that in Xin et al., even for the “high confidence BSJs” (defined as BSJs with absolute read count = 2 across all libraries), the similarity even between technical replicates appears to be relatively modest (see Figure 2a-b in Xin et al.,), which indicates that there may be an issue with rather lenient thresholds for circRNA calling (“noise”).

In summary, we think that estimates such as the “more than 400’000 circRNAs from human samples” are the result of (i) the inclusion of many cancer cell line samples in the databases that contribute with abnormal gene expression and splicing events and of (ii) relaxed thresholds – in some studies a single reported BSJ read in a single sample is sufficient for entry into the database. We are not convinced that these (in part stochastically occurring?) BSJs are suitable for downstream analyses like those that we carry out in our study.

3) As mentioned in the above paragraphs and shown in Supplementary File 2, we actually do obtain relatively high numbers of BSJs/circRNAs per library (i.e. in the tens of thousands); why, then, do we base our downstream analyses nevertheless only on a few thousand circRNAs?

The reason lies in various filtering steps, which are described in Materials and Methods and also shown in Supplementary File 2 for the example of the human libraries. For human liver, cerebellum and testis we thus begin with approximately 25,000, 50,000 and 40,000 detected BSJs, respectively, per tissue sample (=per biological replicate). After removing poorly mapped events and merging across biological replicates and RNase-treated and -untreated samples, these correspond to a total number of 66,405 (liver), 137,615 (cerebellum) and 94,831 (testis) unique BSJs (RNase-treated and -untreated combined). Of the various filtering steps indicated in Supplementary File 2, the ones that decrease the numbers most strongly are the following: Filtering step 2 leads to a reduction by about 75%; briefly, in this step we remove those circRNAs that can be detected in RNase R-treated samples, but not at all in any of the untreated ones. The rationale behind this filtering step is to remove the very low expression BSJs that would be considered “absent” if only based on RNase R-untreated RNA samples. In Filtering step 3 we analyse enrichment of BSJs in RNase R-treated vs. untreated samples; we remove the BSJs for which the log2-enrichment of treated vs. untreated libraries is less than 1.5. This step removes around 15% of the original read count. Finally, in Filtering step 4, we calculate the mean RPM value for each BSJ across untreated replicate libraries, and keep the BSJs with an RPM > 0.05. We consider these loci strong circRNA candidates and use them for our subsequent analyses.

Taken together, we think we understand the Reviewers’ concern that “it is not clear why the authors only used their own circRNA annotation for the current study, as this might lead to a significant bias in the downstream analysis (e.g. affecting the parental genes of circRNAs to be compared to genes that do not generate circRNAs)”. However, we would like to argue that our approach is actually suited to precisely reduce bias that could occur when combining other studies’ circRNA annotations where available (i.e. using circRNAs that were discovered in other human or mouse samples/studies), and compare them to our data from the more unconventional species such as opossum or macaque, for which only little data is publicly available. Second, our filtering pipeline is designed to provide a pool of robust parental genes and circRNAs rather than a collection of inconsistently or sporadically expressed circRNAs (e.g. circRNAs that may be specific only to some cancer cell lines).

Supplementary File 2 has been newly added to the manuscript; in addition, a new section in Materials and methods, entitled “Generation of high confidence circRNA candidates from the comparison of RNaseR-treated vs. -untreated samples” now explains in detail the various filtering steps.

**Author response table 1. resptable1:** Analysis of circRNA databases. Evolutionary analyses are sensitive and prone to the amplification of noise. We thus used a comprehensive dataset including samples from wild-type tissues and the same sex. In addition, as pointed out by Reviewer 2 “without a mock control, [it] is impossible […] to determine the actual resistance to the degradation by the exonuclease and hence determine which potential backsplicing junctions are really coming from circular molecules.” As most of the samples reported in the public database are not RNase R-treated, none of them could provide sufficiently trustworthy circRNA annotations for our study.

	Species	Sample types, circRNA calling	Expression
circAtlas Wu *et al*. Genome Biol 2020 PMID: 32345360	Human, mouse, rhesus, rat, pig, chicken	1,070 RNA-seq samples collected from 19 normal tissues Identification with CIRI2, DCC, find_circ, CIRCexplorer2 No RNase R-treated samples, but circRNAs need to be detected (1) by at least two of the tools and (2) junctions be covered with a minimum of two independent reads (-> identification on a per sample basis) CircRNA numbers shown in the database for each tissue are pooled, meaning there are for example 225’000 circRNAs in 39 human brain samples -> no separation by region, or minimum overlap (e.g in half of the samples analysed)	No comprehensive expression data provided (i.e. only numerical values for the top 30 circRNAs). For cases where biological samples are pooled (same tissue), the maximal expression value across all samples is used to define high expressors. However, due to strong heterogeneity in expression levels, mean values are frequently magnitudes lower.
Comment: The circRNAs reported are not consistently expressed in the tissue. For the brain, it is for example sufficient for a circRNA to be detected in one out of the 39 brain samples, leading to the high numbers reported. This is reflected in the expression variation observed within the same tissue samples. The authors report the maximal observed value, but the tissue mean seems to be at least a magnitude lower. In addition, no RNase R-treated samples were used for validation. We considered this database inappropriate for our study due to the lack of RNase R-treated expression data, which we consider a more important indicator for a circRNA than the detection by two independent identification methods. Note also statement from publication: “Using this method, we found that the vast majority of circRNAs (an average of 61.7%) could be detected only in one species, with only 797 circRNAs shared by all species, in agreement with previous reports on the highly species-specific expression of circRNAs”
circBase Glažar *et al.* RNA 2014 PMID: 25234927	Human, mouse, worm, fly	Mostly cell lines. Mouse cerebellum is the only tissue comparable to our study. The authors report 2,407 circRNAs for this tissue (but: only one cerebellum sample was used in study).	Ranking of circRNAs according to “Scores” that are difficult to transform to expression levels.
Comment: Due to the high number of cell lines samples, we did not consider this database. In addition, only human and mouse samples are present, yet we were looking for an evolutionary dataset that included multiple mammalian species.
CIRCpedia Zhang *et al.* Genome Res 2016 PMID: 27365365	Human, mouse, rat, zebra fish, fly, worm	180 RNA-seq datasets (at first sight, mostly cell lines). One detection tool used. Only 20 out of the 180 samples are RNase Rtreated samples. 4 out of the 20 RNase Rtreated samples are from a mammalian species.	
Comment: This database was not used because it mainly contains cell line samples and only a very low number of RNase R-treated samples.

2. A complementary question is whether the model put forward by the authors applies to highly abundant circRNAs, which are likely to be enriched in functional circRNAs, as the sense in the field is that only highly abundant circRNAs -particularly in the nervous system- might be functional. Can the authors partition their assessment of conservation for the different quantiles of expression? It is indeed possible that only circRNAs in the top quantile of expression will provide insights into the potential functional importance of circRNAs. The authors could also take the opposite approach: for those few circRNAs for which some functionality has been proven, are the pattern of evolution and elements flanking the circularizable exons similar to the ones they describe generally?

We thank the Reviewers for this excellent suggestion. We have taken the proposed approach and compared properties of the top 10% most highly expressed circRNAs with the lower 90%. We have included these interesting analyses in the revised manuscript – mainly in Supplementary File 9, with further information in Supplementary File 10 and Figure 3—figure supplement 5.

First, we compared the general properties of the highly expressed vs. other circRNAs. In short, this analysis revealed that highly expressed circRNA are more likely to be expressed in all three tissues of a species, to be generated from a hotspot, and to be shared between species. Second, we went to the generalised linear model (Table 1) that we used in our study to distinguish parental vs. non-parental genes. Within the model, we compared the prediction values for the highly vs. lowly expressed circRNAs. For opossum, rhesus macaque and human, we found that the highly expressed parental genes were associated with higher prediction values. This effect was mainly driven by GC content (opossum, rhesus macaque, human) and genomic length (rhesus macaque, human).

In summary, our analyses therefore suggest that the overall model that we put forward in our study also applies to highly expressed circRNAs. The overall higher shared-ness of this group of circRNAs makes it probable that potentially functional circRNAs are more likely to be found in this group as well. These findings have been integrated in the manuscript.

Finally, we have analysed known functional circRNAs in our model – please see Figure 3-Figure supplement 3, “Properties of ‘functional circRNAs’ from literature.”

3. The authors identified circRNAs using RNAseR-treated samples. While RNAseR enriches generally for circRNAs, the authors don't have data reporting the abundance of the circRNAs in mock treated samples. Without a mock control is impossible for the authors to determine the actual resistance to the degradation by the exonuclease and hence determine which potential backsplicing junctions are really coming from circular molecules. The authors could do at least for some of the tissues/species show that the circRNAs they identified have been previously shown to be real (e.g., by a canonical comparison mock vs RNAseR treated samples).

We apologize for not having described more explicitly the experimental protocol for library generation in the main text, but only in the Materials and methods section and in the Supplemental Material. We fully agree with the Reviewers’ assessment that to conclude actual resistance to RNase R treatment (and thus, likely, circularity), a mock control is required. All RNase R-treated libraries of our study were accompanied by a parallel library from the same RNA preparation, but without prior RNase R treatment. In our previously submitted version, this information could be found, for example, in Figure 1—figure supplement 1. Moreover, we had described the use of the mock-treated samples in the materials as methods in two separate paragraphs. We therefore feel that this “Essential Revision” item does not require additional analyses (or experiments), but rather an improvement in presentation in the main text, in order to avoid misunderstandings. We have thus adapted the manuscript now giving all essential information at the beginning of the Results section. We have also moved the figure panel that shows the RNase R treatment vs. control strategy from Figure 1—figure supplement 1 to Figure 1 to show the protocol details more prominently. Finally, in the Materials and methods section, we have expanded the explanation of the various steps for filtering and RNase R enrichment analyses – see new section entitled “Generation of high confidence circRNA candidates from the comparison of RNase R-treated vs. -untreated samples”.

4. For the analysis of enriched TEs of flanking introns of circRNAs or circRNAs shared across species, except the ages of TEs, other factors should be also included in these analyses. For example, whether pairing abilities of RVCs alone can explain the phenomenon? Whether the number of RVCs in the flanking introns is related to the conserved or species specific circRNAs? In addition, In this case, the general conclusion about circRNA biogenesis and TE insertion events can be biased.

We thank the Reviewers for the opportunity to clarify these issues. To address them, we have conducted a more refined analysis of the dimer composition in the flanking introns of shared and species-specific circRNA loci. The obtained results have led to a change in the presentation and layout of the final part of the Results section (“Flanking introns of shared circRNA loci are enriched in evolutionarily young TEs”) and of Figure 5.

We would like to point out that our comparison of TEs in flanking introns in shared and species-specific circRNAs was – already in the first version of the manuscript – based on multiple variables, notably: the phylogenetic distance (=age) of the TE, the number of observed dimers (=frequency), their degradation rates (=milliDiv), as well as the binding affinity of a given dimer (=minimal free energy, MFE). However, after having received the Reviewers’ comments and upon reexamining our original manuscript version, we realised that we had indeed not made this sufficiently clear in text and figures.

We have therefore made the following changes. First, we already clarified some of the conceptual points in Figure 4 (where we analyse parental vs. non-parental genes). We now make it clearer where and how age and binding affinity are assessed in Figure 4C-F (mouse) and Figure 4—figure supplement 2 (other species). In particular, it should now be clearer that Figure 4D refers to age and Figure 4E to binding energy. The former use of the phrase “binding score” was changed in the new version to “TE pairing score (age + MFE)”, to indicate that it integrates both the components age and binding energy (Figure 4F).

The major changes relating to Essential revision 4 we implemented in Figure 5, where shared vs. species-specific circRNAs are compared. Current Figure 5A and Figure 5-Figure supplement 1 now show the dimer count per gene on the x- and y-axis (the corresponding panel in the previous manuscript version was based on total count); the “per gene” analysis facilitates the interpretation of enrichment in shared vs. species-specific dimer counts. Furthermore, we have included a new supplementary figure – Figure 5—figure supplement 2 – to portray the increase of possible dimers that can be formed (“dimer interaction landscape”) due to increased number of dimers in the flanking introns of shared vs. species specific circRNA loci.

Next, we reconsidered how we could calculate more precisely the actual degradation rates of the relevant dimers; the outcome is found in the new Figure 5C and Figure 5-Figure supplement 3 (left panel). Briefly, we now show a comparison of milliDiv values between the individual predicted dimer-forming repeats – rather than the mean milliDiv values from repeats of the same dimer, as in the old version – should conceptually be a more precise approach. With this analysis, we now conclude that degradation levels between top-5 dimers do not differ between the flanking introns of shared and species-specific circRNA loci. Because degradation rates alone may not fully capture the actual decline in pairing of the dimer, we next added the analysis of the actual binding affinity for all top-5 dimers (Figure 5, right panel) calculated from the individual genomic sequences (rather than from the reference sequence). In the next analysis, we selected for each gene the least degraded dimer, i.e. the one that is strongest and that is likely to form (see Material and Methods). Again these analyses showed no significant difference between the flanking introns of shared and species-specific circRNA loci, fully in line with our model.

We believe that these changes have significantly improved our analysis. However, our deep analyses of dimer landscapes are rather novel; in this context, we would like to note that we are quite aware that dimer formation will be the result of a complex interplay between TE frequency, degradation levels, and sequence affinity – how these individual parameters should be weighted and how they can be integrated into a quantitative measure is not simple to evaluate. Nevertheless, all our analyses suggest that degradation rates and sequence affinities are fairly similar between shared and species-specific circRNA loci; we therefore also assume that TE frequency is one of the driving factors for circRNA formation.

Suggested text revisions:a. The discussion should further elaborate the concept that at least for some circRNAs, their main functionality is to regulate/limit the expression of the host gene, hence the exact exons that form circRNAs might not be important. Therefore it would not so important which exon in particular makes the cirRNA or even which elements in the flanking introns are driving this process but rather the existence of this during evolution. This is relevant especially for those circRNAs for which the circular molecules are abundant in comparison with the linear mRNAs generated from the locus and there are at least some cases in the literature in which has been shown that production of the circRNA comes at the expense of the linear transcript.

We now more clearly include this point towards the end of the discussion.

b. Discuss the evolutionary implications of the observation that circRNAs in fruitfly is not driven by intronic pairing of repetitive/transposable elements (Westholm, et al., Cell Rep 2014, PMID: 25544350).

We have looked at the cited publication in some detail in order to see how we could best accommodate this suggested text revision. First, we noticed that the publication is relatively careful in its wording and does not explicitly state that “circRNAs in fruitfly is not driven by intronic pairing of repetitive/transposable elements”. A telling statement from the paper is, for example: “However, overall, it appears that *Drosophila* RNA circularization does not appear to be driven by flanking sequence or structural complementarity, as in mammals.” - the authors are thus very careful in their wording in order to avoid overstating this particular point. Closer examination of their data shows, for example, in Figure S3A that their approach using MEME recovers one short substring of the ALU element described by Jeck et al., (2012) in human circRNA flanking introns (although, of note, this Figure unfortunately does not show the same analysis in control introns). In Figure S3B, a comparison of *Drosophila* introns (500 nt window) recovers some repetitive sequences that are of lower complexity and found in both circRNA and non-circRNA introns, and that are therefore dismissed as possible transposable element substrings. Still, their Figure S4B actually does show that circRNA flanking introns contain significantly better stem-loop-potential than non-circRNA introns, especially for short sequence window sizes (interestingly, the effect is clear until window sizes of 100 nt, but absent when using a 500 nt window, possibly indicating that the MEME analysis could have resulted in other outcomes when using smaller windows as well). Westholm et al., do not extensively develop the lack of evidence for intron pairing or repetitive/transposable elements in their study, but concentrate on other main findings. As it stands and only based on this study, therefore, we feel it would be misleading to develop and discuss evolutionary scenarios in our mammalian-centric study.

The situation is also further complicated by the fact that Kramer et al., (2015) (PMID: 26450910) actually have suggested a role for intronic repeats in circRNA biogenesis in *Drosophila*. Also Ashwal-Fluss et al., (2014) (PMID 25242144) identify numerous reverse complementary repeats between the two flanking introns of the mbl circRNA (see Figure S2B in their publication). Despite the specific role of MBL in this circularisation event, this finding suggests that sequence-dependent pre-mRNA folding may have an additional, and possibly more ancestral, role in circRNA biogenesis in this particular example (reviewed in Patop et al., (2019) PMID: 31343080). Finally, there is good evidence for reverse-complement sequence mediated circRNA biogenesis in *C. elegans* (Ivanov et al., 2015; PMID 25558066). With all these caveats to be navigated – and with the idea of simplifying/shortening our discussion in view of other reviewers’ comments, we hope the editor and reviewers will accept that we prefer not to dedicate a part of the discussion to conservation of circRNA biogenesis beyond the subject of our mammal-based study.

c. On page 6, the authors stated "Coding genes in rhesus macaque and human are characterised by a bimodal GC content distribution". The H3 category contains all genes with GC content > 52% while L2 category contains genes with 37-42% GC contents (Figure 3A). It is not correct to define it as bimodal GC content distribution when the group standard has a large difference.

This has been changed.

Former sentences:

“Coding genes in rhesus macaque and human are characterised by a bimodal GC content distribution (see peaks in L2 and H3 for non-parental genes). By contrast, the two rodents displayed a unimodal distribution (peak in H1), whereas opossum coding genes were generally GC-poor (in agreement with Galtier and Mouchiroud, 1998; Mikkelsen et al., 2007).”

have thus been changed in the revised version to:

“Non-parental genes displayed a unimodal distribution in the two rodents (peak in H1), were generally GC-poor in opossum (peak in L1), and showed a more complex isochor structure in rhesus macaque and human (peaks in L2 and H3), in agreement with previous findings (Galtier and Mouchiroud, 1998).”

d. More data are needed to sustain the statement "circRNA parental genes are characterised by low GC content and high sequence repetitiveness". Low GC content and high sequence repetitiveness may be the consequences of the long intron length of parental genes and/or the enrichment of repeat elements in long introns. More stringent conditions, like comparable intron lengths between circRNA and non-circRNA parent genes would need to be set up for this type of comparison.

To address this “text revision”, we have now rephrased the cited statement, e.g. in the subsection title (now: “CircRNA parental genes are associated with low GC content and high sequence repetitiveness.”).

In terms of the need for additional data to support the statement, we would like to point out that in particular the modeling approach that we applied actually is designed to isolate the individual predictors, i.e. it calculates the association of, for example, GC content, over the other effects, such as intron length. We are therefore not fully sure what is meant with the comment that “more stringent conditions are needed”.

Reviewer #1 (Recommendations for the authors):I would recommend the authors to consider the following revisions:1. There seems to be some contradiction between the argument that circRNAs tend to be generated from constitutive, widely expressed exons and the statement on page 3 that circRNAs "showed considerable tissue-specificity".

We have now rephrased this sentence to be clearer:

“Identified circRNAs were generally small in size, overlapped with protein-coding exons, frequently detectable only in one of the tissues, and were flanked by long introns (Figure 1—figure supplement 3).”

This wording is better compatible with the main underlying reason for “tissue-specificity”, namely that the stochastic splicing noise around the constitutive exons will not lead to reproducible circle formation across all tissues. Second, similar to linear alternative splicing, one needs to differentiate between the transcript and exon level: alternative RNA isoforms can show tissue-specific expression patterns, while being partially composed of the same (constitutive) exons – the exon itself can be considered expressed in multiple tissues (because it is used in multiple, but different isoforms), yet the transcript is not.

2. The authors may want to elaborate further on the possible links between the preferred genomic architectures of circRNAs hotspots and the concept of exon definition (e.g. enrichment in long flanking introns and differences in GC content between exons and introns, as reported by Ast and colleagues).

We agree that this would be a very interesting idea to address in the future. However, we felt that in its current state the manuscript was already quite loaded with information analyses, including comments that we should shorten the discussion. We therefore did not extend our analyses in this additional direction at this point. We hope the Reviewer will find this decision acceptable.

3. The predictive model generated by the authors produces very impressive results. In addition to sequence complementarity, a number of protein factors have been shown to support the formation of stem-loop structures associated with back splicing, including hnRNP L, MBNL1 or PTB. The authors may be in an excellent position to assess whether the presence of potential binding sites for these proteins contributes evolutionarily to the generation of circRNAs (or subtypes of them). More specifically, does the inclusion of binding sites for these factors improve the performance of their predictive model?

The reviewer raises an excellent point and opportunity for further in-depth analyses.

It is indeed possible that circRNAs become first expressed by the formation of hairpin-like structures driven by TEs in their flanking introns. In a second step (and in case that the circRNA has conferred a benefit for the organism), circRNA expression may be stabilised by the evolution of new binding sites for proteins such as hnRNP proteins, MBNL1 and PTB. At the same time, this reduces the selective pressure on the TE, leading to its degradation and circRNA candidates without detectable dimers. While we focus on the “first steps” towards the expression of a circRNA, the integration of binding site information from external datasets to understand “what happens next” can be considered an excellent starting point for a follow up study. Yes, as for the previous point, we found this interesting request too complex to accommodate at this point, and thus beyond the scope of this first study. We do, however, expand a bit more on RBPs in the discussion.

4. Figure 4A: it would be good to provide some measurement of the statistical relevance of the enrichment scores observed.

Measurements of statistical relevance have been integrated in the figure and the statistical tests used are mentioned in the figure legend.

5. Figure 5 and related discussion about evolutionary age of TEs associated with circRNA biogenesis: is it possible that dimers co-evolving over long periods of time accumulate mutations but retain -through compensatory changes- their capacity to form base pairing interactions leading to looping out and back-splicing?

The Reviewer raises an interesting point. Our modified Figure 5 (see above, Essential revision 4) now looks at aspects of this idea by calculating the actual binding energies of the local TE copies predicted to form dimers.

In general, however, we would think that most sequence changes – even the compensatory ones – are rather random in nature than selected for. The most relevant TEs that we identify are SINE elements, which are rather GC-rich, but integrate into GC-poor environments. Our analyses (not shown in the manuscript) indicate that many of the mutations we observe are G/C -> A/T changes, for which we would tend to hypothesize that they occur as a consequence of the GC-poor environment to which the integrated TE is adapting. This GC adaptation to the local GC content will happen in both TEs of the dimer in parallel, which can lead on average to more seemingly “compensatory” mutations than expected. However, these compensatory changes would not be driven by selective pressure, but by the local GC content. Therefore, although it is possible that a few of them are real compensatory mutations, it would represent a major project to properly identify, classify and analyse these signals.

Reviewer #2 (Recommendations for the authors):[…]The regulation of circRNA biogenesis can be complex. It should be careful to evaluate different conditions to draw a convincing conclusion. The major pitfall of this study is to use relatively small population of circRNAs for the evolutionary study, which may lead to biased findings. In addition, more stringent controls are required for analyses throughout the study. The Discussion part seems to be unusually long, containing lots of assumptions that are needed to be proven. Please shorten.

The first part of Reviewer #2’s recommendations (“The major pitfall of this study is to use relatively small population of circRNAs […]”) is largely overlapping with Essential revision 1 at the beginning of this rebuttal document. Please refer to this section for a detailed response.

The context of the second statement “In addition, more stringent controls are required for analyses throughout the study.“ is not fully clear from the above paragraph. However, in the more detailed “Public evaluation summary”, we find as point 4 the statement: “More data are needed for the statement "circRNA parental genes are characterised by low GC content and high sequence repetitiveness". Low GC content and high sequence repetitiveness may be the consequences of the long intron length of parental genes and/or the enrichment of repeat elements in long introns. More stringent conditions, like comparable intron lengths between circRNA and non-circRNA parent genes, should be set up for this type of comparison.”

We agree with the Reviewer that it is important to account for a possible dependence of variables, as shown by the example that the Reviewer cites: GC content and sequence repetitiveness will clearly not be independent from intron length. This is precisely why we used generalised linear models, which is the state-of-the-art approach when multiple correlated dependent variables are predicted. To avoid possible biases by correlated variables, we have also assessed all variables for multicollinearity before including them into the analysis (see Material and Methods, section Generalised linear models). We prefer this approach of a global model that includes all potential predictive parameters, over the Reviewer’s suggestion of individual comparisons of single features, precisely because such comparisons are prone to biases like the one described by the Reviewer.

The third point – shortening of discussion and removal of assumptions – is well taken. We have substantially restructured and rewritten the discussion accordingly.

Reviewer #3 (Recommendations for the authors):[…]The manuscript is interesting and timely. However, the conclusions are too broad and the specific issues need to be addressed experimentally and/or by doing additional analysis.More specifically:– The approach for circRNA detection is at least problematic. Without a mock control is impossible for the authors to determine the actual resistance to the degradation by the exonuclease and hence determine which potential backsplicing junctions are really coming from circular molecules. One thing the authors could do is at least for some of the tissues/species show that the circRNAs they identified have been previously showed to be real (e.g., by a canonical comparison mock vs RNAseR treated samples).

This point has been answered under Essential Revision 3.

– Authors seems to ignore that in vivo studies focused on highly expressed circRNAs. Can the authors partition their assessment of conservation for the different quantiles of expression? Indeed, I would believe that only circRNAs in the top quantile of expression are the ones that will really give insights into the potential functional importance of circRNAs.

This point has been answered under Essential Revision 2.

– Notion of noise for lowly expressed circRNAs is prevalent in the field and the key question is whether the circRNAs expressed in the neural system at high levels have function. So only on those circRNAs is relevant to determine evolutionary conservation. Indeed, the authors could take the opposite approach; for those few circRNAs for which some functionality been proven, are the pattern of evolution and elements flanking the circularizable exons similar to the ones they describe generally?

The largest part of this point, too, is picked up by several of the Essential revisions - please refer to the beginning of this document. Moreover, please see Figure 3-Figure supplement 3 for an analysis of known, functional circRNAs.

– The manuscript ignores potential cis regulatory mechanisms of circRNA biogenesis. One could argue that an important feature of a given locus is to produce circRNAs as a way to limit the amount of linear RNA. In this sense, it would not so important which exon in particular makes the cirRNA or even which elements in the flanking introns are driving this process but rather the existence of this during elevation. This is relevant especially for those circRNAs for which the circular molecules are abundant in comparison with the linear mRNAs generated from the locus and there are at least some cases in the literature in which has been shown that production of the circRNA comes at the expense of the linear transcript. The authors could include this also as a criterion for evolution and not rule it out based on some theoretical assumptions.

We now dedicate more space to this important concept in the discussion of the revised manuscript (see also above our response to suggested text revisions).